# Testing the validity of regional detail in global analyses of Sea surface temperature — the case of Chinese coastal waters

Yan Li[1][*], Hans von Storch [2,3], Qingyuan Wang [4], Qingliang Zhou [5], Shengquan Tang[2,3]

[1]National Marine Data and Information Service, Tianjin, People's Republic of China

[2]Institut für Küstenforschung, Helmholtz Zentrum Geesthacht, Germany

[3]Ocean University of China, Qingdao, People's Republic of China

[4]Tianjin Meteorological Observatory, Tianjin, People's Republic of China

[5]Chinese Meteorological Administration, Beijing, People's Republic of China

**Abstract.** We have designed a method for testing the quality of multidecadal analyses of SST in regional seas by using a set of high-quality local SST observations. In recognizing that local data may reflect local effects, we focus on dominant EOFs of the local data and of the localized data of the gridded SST analyses. We examine the patterns, variability as well as trends of the principal components. This method is applied to examine three different SST analyses, i.e., HadISST1, ERSST and COBE SST. They have been assessed using a newly constructed high-quality data set of SST at 26 coastal stations along the Chinese coast in 1960–2015 which underwent careful examination with respect to quality, and a number of corrections of inhomogeneities. The three gridded analyses perform by and large well from 1960 to 2015, in particular since 1980. However, for the pre-satellite period, prior to 1980s, the analyses differ among each other and show some inconsistencies with the local data, such as artificial break points, periods of bias and differences in trends. We conclude that gridded SST-analyses need improvement in the pre-satellite period (prior to 1980s), by re-examining in detail archives of local quality-controlled SST data in many data-sparse regions of the world.

[*] Corresponding author. E-mail address: ly_nmdis@163.com

## 1. Introduction

Sea surface temperature (SST) is a key parameter for climate change assessments. It is significantly associated with many atmospheric and oceanographic modes, such as Pacific Decadal Oscillation (PDO), El Niño/South Oscillation (ENSO), Indian Ocean Dipole (IOD), etc. (Saji et al., 1999, Mantua and Hare, 2002, Yeh and Kim, 2010). Long-term historical SST data sets have been extensively used as a source of information on global and regional SST trends and variability (Belkin, 2009; Wu et al., 2012; Boehme et al.2014; Hirahara et al.2014; Stramska and Bialogrodzka, 2015). However, historical SST datasets have large uncertainties in long-term trend patterns in some regions. For example, observed SST changes in the tropical Pacific are still controversial, depending on the choice of the dataset and study period (Bunge & Clarke 2009). Vecchiga et al. (2008) indicated that the equatorial zonal SST gradient in the Pacific has intensified in Hadley Centre Sea Ice and Sea Surface Temperature (HadISST) but weakened in Extended Reconstructed SST (ERSST) from the nineteenth to twentieth centuries. Scientists utilized several different datasets, including the reconstructed and un-interpolated datasets, to study the SST variability in tropical areas and the China Seas (Xie et al., 2010; Liu and Zhang 2013, Tokinaga et al., 2012). They found that there were large uncertainties in estimate of SST warming patterns using different SST datasets. Thus, it is also necessary for comparing different SST products over the regional areas in detail.

Coastal marine ecosystems yield nearly half of the earth's total ecosystem goods and services (Costanza, 1997). A study of SST changes in the world ocean with large marine ecosystems revealed that the Subarctic Gyre, European Seas, and East Asian Seas warmed at rates 2–4 times the global mean rate (Belkin 2009). Recently, Lima and Wethey 2012 using a SST dataset with higher spatial-temporal resolution detected that during the last three decades ~ 71.6% of the world coastal locations have experienced a warming trend of 0.25±0.13℃ per decade and 6.8% a cooling of -0.11±0.10℃ per decade. Increase in SST is especially important in coastal areas due to its strong impact in coastal ecosystems (Honkoop et al., 1998; Burrow et al., 2011; Wernberg et al. 2016). Simultaneously, coastal SST is highly influenced by local factors, such as the anthropogenic land-based processes, upwelling currents, fresh water discharge, ocean fronts and local tidal mixing. An accurate analysis of the local SST and its variability is needed for marine ecosystem-based management. Here, we mainly focus on three globally gridded SST datasets, that is, the HadISST1, ERSST, COBE SST (Rayner et al., 2003, Ishii et al., 2005, Smith et al., 2008, Hirahara et al., 2014; Huang et al., 2015). Besides, a fourth SST product is considered, i.e., NOAA Optimum Interpolation SST (OISST)

version 2 using Advanced Very High Resolution Radiometer infrared satellite SST data from the Pathfinder satellite combined with buoy data, ship data, and sea ice data, covering from 1982 to present. Because of its high spatial resolution of 0.25 °×0.25 °, it is used in the concluding section for clarifying some additional aspect. All of these datasets have been widely used in the regional and global climate change studies. Given that these datasets have been developed by independent groups, there are some differences of data sources, bias adjustment and reconstruction method, etc. in the SST analyses products. For example, some analyses only use in situ observations, such as ERSST v4 and COBE SST. Others use both in situ and satellite observations, such as OISST and HadISST1. There are also some differences from quality control and gap-filling choices when and where observations are sparse, particularly in early record periods and coastal areas (Huang et al., 2015; Li et al., 2017). These differences also indicate some uncertainties in these SST analyses. In order to test the validity of these gridded SST datasets along the coast of China, SST records for the period of 1960–2015 at total 26 Chinese coastal hydrological stations coast are used. All of these *in situ* SST data from 1960 to 2015 are provided by the National Marine Data and Information Service (NMDIS) of China and have been quality controlled and homogenized by Li et al. (2018). These SST data from coastal hydrological stations have never been merged into HadISST, COBE SST or other gridded SST analyses. Therefore, the homogenized long-term SST observations along the Chinese coast can be used for evaluation on these analyses. We study the performance of these gridded SST datasets in the coastal waters by comparing to the homogenized SST.

Thus, the remainder of this paper is structured as follows: Details on the observational and gridded data sets and methodology used in this study are given in section 2. Section 3 introduces the local homogenized SST series along the Chinese coast (Li et al., 2018), which is used as a reference to compare to the gridded data sets with. For adding confidence in the quality of this local SST data set, these SST data are compared with an independently constructed local air temperature data. The basic statistics of the local SST-data series are also shown. Section 4 describes the results and comparisons with gridded SST data sets in the Chinese coastal waters. Further discussion and conclusion are given in section 5.

## 2. Data and methodology

### 2.1. Data source

The SST records during 1960–2015 at the 26 sites of coastal hydrological stations along the Chinese coast have been assembled and homogenized. Homogenized monthly mean surface

air temperature (SAT) series from National Meteorological Information Center (NMIC) of China (Xu et al., 2013) and the gridded SAT from the latest version of the Climate Research Unit's (CRU) gridded high resolution (0.5°×0.5°) dataset CRU TS 3.24.01 for 1960–2015 (Harris et al., 2014) are used to investigate the consistency of homogenized SST data with the local SAT.

Four globally gridded SST datasets are used in our work (see Table 1): (1) The $1°×1°$ Hadley Center Sea Ice and Sea Surface Temperature monthly dataset (HadISST) (Rayner et al., 2003); (2) The $1°×1°$ Centennial In Situ Observation-Based Estimates of the Variability of SST (COBE SST) (Hirahara et al., 2014); (3) $2°×2°$ Extended Reconstructed Sea Surface Temperature version 4 (ERSST v4) for 1960–2015 (Smith et al., 2008, Huang et al., 2015). (4) NOAA OISST v2 for 1982–2015 (Reynolds et al. 2007).

**Table 1**. Global gridded SST datasets that are used in this study

| Dataset | Resolution | Period | Sources |
|---------|-----------|--------|---------|
| ERSST v4 | 2°x 2° | 1960–2015 | http://www.ncdc.noaa.gov/oa/climate/research/sst/ERSST. v4.php |
| HadISST | 1°x 1° | 1960–2015 | http://www.metoffice.gov.uk/hadobs/hadisst/data/download.html |
| COBE SST | 1°x 1° | 1960–2015 | http://ds.data.jma.go.jp/tcc/tcc/products/elnino/cobesst/cobe-sst.html |
| OISST | $\frac{1}{4}°x\frac{1}{4}°$ | 1982–2015 | http://www.ncdc.noaa.gov/oisst |

## 2.2. Methodology

Statistical methods such as conventional empirical orthogonal function (EOF) (Kim et al., 1996, von Storch and Zwiers 1999), correlation analysis and linear trend analysis are employed. The significance of each trend has been tested by the Mann-Kendall test using Sen's slope estimates quantify trends (Sen, 1968). The tests were stipulated to operate with a probability for a false rejection of the null hypotheses (i.e., zero trend) of 5%. They are conducted with the implicit assumption that the data are serially independent. There are only weakly correlated but not really independent. Thus, the tests are "liberal", i.e., have tendencies for falsely rejecting too often the null hypothesis, when it is actually valid (von Storch and Zwiers, 1999). However, since the effect is relatively weak, given the small serial correlations, and since we have no results, which are close to the stipulated critical levels, we

do as if the serial dependence is not of importance. However, this caveat should be kept in mind, when assessing the results.

## 3.  The local homogenized SST records along the Chinese coast

Currently, more than 100 coastal hydrological stations are operating and monitoring near-shore hydrological conditions. Among these stations, only 26 stations have routinely and continuously recorded since 1960, with a percentage of missing data less than 4%, Also, these stations have undergone only a few (five or less) documented relocations. The locations of the 26 coastal hydrological stations are shown in Fig.1a. Due to the fact that this area between 29 N (Station 11) and 35 N (Station 10) is a vast muddy coast which is not suitable for hydrological stations, there are only 10 hydrological stations. Among them, some stations were built up after 2000s and some have much missing data. That is why no station has been chosen between 29 N (Station 11) and 35 N (Station 10). Monthly mean SST series were then derived and subjected to a statistical homogeneity test, called the Penalized Maximum T (PMT) test (more details can be found in Li et al., 2018). Homogenized monthly mean SST series were obtained by adjusting all significant change points which were supported by historic metadata information. These identified change points at each station are displayed in Fig.1b. The majority of change points are caused by instrument changes and station relocations, accounting for 60.6% and 24.6% of the total, respectively. In our work, we consider annual mean values. Some analyses with seasonal mean values are also calculated, but these are not covered by our present account and merely summarized. The supporting evidences are provided by the Supplementary Online Material (SOM) in Appendix B.

The standard statistics derived from the data in the period of 1960–2015, that is, long-term mean, the standard deviation of annual means and the decadal trends are listed in Table 2. SSTs vary along the Chinese coast, between about 11.5 ℃ at the north and 25 ℃ at the southernmost locations. The standard deviations are of the order of 0.50 ℃ at all locations, with a maximum of 0.71 ℃ and a minimum of 0.43 ℃. The decadal trends vary between 0.13 ℃ per decade to 0.29 ℃ per decade. Table 2 also provides the long-term means of the homogenized data and of the raw (unhomogenized) data. The differences between the homogenized data and the raw data (last column) vary between -2.26 K and 0.53 K. At 22 of the 26 stations, a downward correction of the mean has been found necessary – only at Station 15 (Pingtan) and Station 23 (Weizhou) an upward change was stipulated, and in two case nearly no change of the mean at Station 7 (Shidao) and Station 24 (Naozhou).


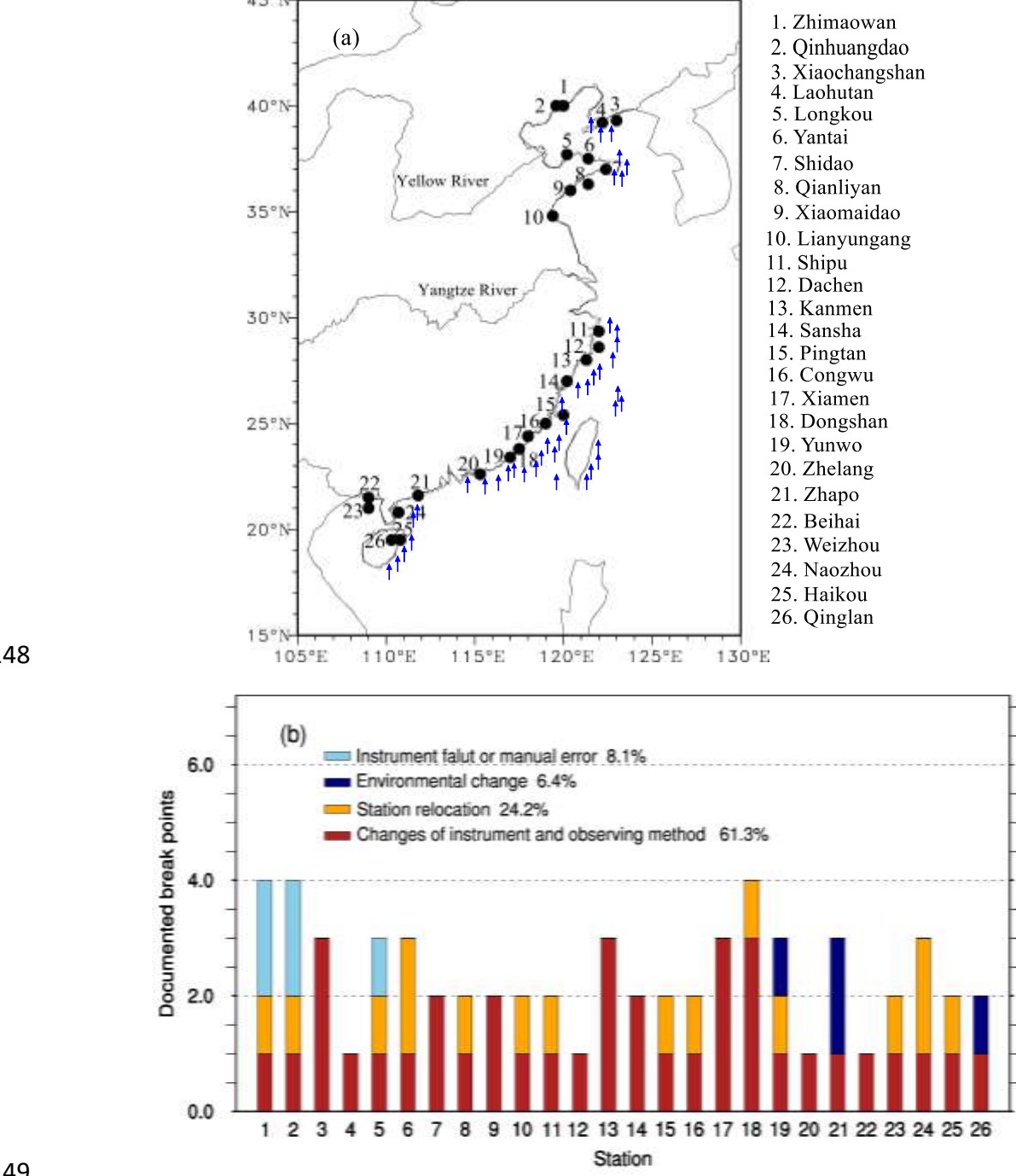

1. Zhimaowan
2. Qinhuangdao
3. Xiaochangshan
4. Laohutan
5. Longkou
6. Yantai
7. Shidao
8. Qianliyan
9. Xiaomaidao
10. Lianyungang
11. Shipu
12. Dachen
13. Kanmen
14. Sansha
15. Pingtan
16. Congwu
17. Xiamen
18. Dongshan
19. Yunwo
20. Zhelang
21. Zhapo
22. Beihai
23. Weizhou
24. Naozhou
25. Haikou
26. Qinglan



**Figure 1**. Study area and locations of 26 coastal sites (a), for which continuous monthly SST recordings are
available and corrected by eliminating inhomogeneities. The number of identified breakpoints in individual
SST stations from 1960–2015 (b). Result from Li et al. (2018). Black circle represents 26 coastal sites and
blue arrow represents coastal upwelling.

**Table 2.** Statistics of the time series of the annual homogenized local SST, plus the differences to the raw
data, which were used to construct the homogenized series (columns 6 and 7).

| Station No. | Full name | Mean homogenized SST | Standard deviation | Trend (℃/10yrs) | Mean unhomogenized SST | Diff |
|---|---|---|---|---|---|---|
| 1 | Zhimaowan | 11.50 | 0.53 | 0.17 | 11.75 | -0.25 |
| 2 | Qinhuangdao | 12.21 | 0.59 | 0.26 | 12.32 | -0.11 |
| 3 | Xiaochangshan | 11.54 | 0.71 | 0.29 | 11.73 | -0.19 |
| 4 | Laohutan | 11.36 | 0.59 | 0.21 | 11.47 | -0.11 |
| 5 | Longkou | 13.36 | 0.59 | 0.22 | 13.51 | -0.15 |
| 6 | Yantai | 12.65 | 0.59 | 0.17 | 12.79 | -0.14 |
| 7 | Shidao | 12.09 | 0.59 | 0.14 | 12.08 | 0.01 |
| 8 | Qianliyan | 14.37 | 0.65 | 0.17 | 14.41 | -0.04 |
| 9 | Xiaomaidao | 13.76 | 0.63 | 0.22 | 13.84 | -0.08 |
| 10 | Lianyungang | 14.85 | 0.57 | 0.21 | 14.94 | -0.08 |
| 11 | Shipu | 17.41 | 0.65 | 0.26 | 18.01 | -0.61 |
| 12 | Dachen | 17.67 | 0.65 | 0.24 | 17.91 | -0.24 |
| 13 | Kanmen | 18.20 | 0.56 | 0.17 | 18.42 | -0.22 |
| 14 | Sansha | 19.21 | 0.71 | 0.21 | 19.91 | -0.19 |
| 15 | Pingtan | 19.72 | 0.61 | 0.19 | 19.45 | 0.53 |
| 16 | Congwu | 19.98 | 0.52 | 0.17 | 22.18 | -0.64 |
| 17 | Xiamen | 21.50 | 0.51 | 0.19 | 21.47 | -2.26 |
| 18 | Dongshan | 20.84 | 0.45 | 0.13 | 21.12 | -0.28 |
| 19 | Yunwo | 21.02 | 0.44 | 0.13 | 21.36 | -0.34 |
| 20 | Zhelang | 22.43 | 0.44 | 0.15 | 22.62 | -0.19 |
| 21 | Zhapo | 23.62 | 0.50 | 0.18 | 23.68 | -0.06 |
| 22 | Beihai | 23.60 | 0.55 | 0.18 | 24.06 | -0.46 |
| 23 | Weizhou | 25.79 | 0.43 | 0.17 | 25.66 | 0.13 |
| 24 | Naozhou | 24.46 | 0.49 | 0.16 | 24.44 | 0.02 |
| 25 | Haikou | 25.00 | 0.49 | 0.16 | 25.10 | -0.10 |
| 26 | Qinglan | 25.80 | 0.44 | 0.18 | 25.86 | -0.07 |

The quality of the data set has already been documented by Li et al. (2018). To add confidence in the quality of this data set, we compared the new data set to an independent data set of local SAT at 26 nearby local stations. Also, this data set has been homogenized – independently of the processing of the SST series. SST and SAT data are not compares directly pairwise, but in terms of the patterns and coefficient time series (PCs) of their EOFs. The similarity of the principal components is striking. The first PCs share a correlation coefficient of 0.97, and the second 0.86 (Fig.A1). Thus, the SST series are fully consistent with these SAT series. When this exercise is repeated with CRU TS 3.24.01 instead of the in-

situ SAT series, we find a similar consistency (see Fig. SOM-1). The PCs of SAT-CRU also
show high correlations of 0.94 and 0.83 with the in situ SST (see Fig. SOM-1) (more details
are shown in Appendix A and B). Thus, we conclude that our homogenized SST data is
superior to earlier used data on the SST variability and trends along the Chinese coast.

**4. Comparison with gridded SST datasets in the Chinese coast waters**

Given the consistency of the newly homogenized SST series with independent regional SAT
data, we use it as a benchmark for assessing the regional quality of the four globally gridded
SST data sets in Table 1. In the following, we name the new data set "Local homogenized
SST" as "LH", while the datasets extracted from the gridded SST datasets as "localized
analysis data", and use the abbreviation "LA". For instance, LA-HadISST is the SST found in
HadISST in the local grid box, which contains the locations in the LH data set.
These "localized" time series (LA) of the three gridded datasets, which extend to the full time
window 1960–2015 (ERSST, HadISST, COBE SST; referred as LA-ERSST, LA-HadISST,
LA-COBE SST) are then compared to the local series — LH, by first comparing the standard
deviations and the trends, and by calculating from trends, differences (Diff) and the root mean
square errors (RMSEs) for the 26 stations (Table 3). We do this for annual mean values. The
fourth dataset, OISST data, covers a shorter time window from 1982–2015 and has a high
spatial resolution. It is used in the concluding section for clarifying some additional aspects in
the section 5.
For summarizing the results, we compute EOFs of the LH and the LAs, as well as for the
differences of LH and LAs. The LH data are derived from observational stations, whereas the
LA data are representing area values averaged across a grid box. Therefore, the LA data
should vary less than the LH data. Possible mismatches between the local LH data and the
spatial averages of grid box data in the LAs may be related to small scale effects; however,
the usage of EOFs is expected to reduce these truly local specifics, as the first EOFs describe
joint co-variations among the 26 elements in both LA and LH data sets.

**4.1. Comparing with HadISST**

The 56-year mean values of local SST in the analysis LA-HadISST are in all cases higher than
at the local stations (Table 3). Some differences are of the order of 2K and even 3K, in
particular along the East China Sea extending from Station 11 (Shipu) to Station 20 (Zhelang).
To some extent, this difference may reflect differences between averages of a larger coastal
ocean area and *in situ* observations, but not entirely.
The variations in LA are similar to LH, but there are some differences: as expected, 65.4% of
the standard deviations (17) are larger for LH, and 34.6% cases (9) smaller. The correlations
are all large enough to reject the null hypothesis of the absence of a link (if we assume serially
independence the 90%-critical value is 0.22) except for the northernmost Station 19 (Yunwo).
Part of the difference to the ideal value of 1 may be due to the different spatial scale, but
values as low as 0.41 indicate to more systematic differences. The trends are positive for all
sites (Table 3) – only the northernmost Station 1 (Zhimaowan) signals a weak downward
trend in the LA-HadISST data set. In about 50% of the case, the coastal sea warms faster
according to LH than to LA-HadISST, and for 50% it is the opposite. For the two
northernmost sites, Station 1 (Zhimaowan) and Station 2 (Qinhuangdao), the warming
according to LA is very weak, whereas along the stretch from Station 15 (Pingtan) to Station
19 (Yunwo) the warming according to LA-HadISST is considerably stronger than in LH.
The time series for the two northern sites in the Bohai Sea are shown in Fig. 2. The sequence
of maxima and minima share some similarity, but the trends differ markedly. The LH curves
(red lines) exhibit both a steady increase, whereas the LA-HadISST curves (black lines) tend
to decline in the first 10–20 years, and to vary at a mostly constant level (Fig.2a and 2b). In
this case, the "story told" by LH is considerably different than that of LA-HadISST.
The time series of the SST averaged across the stations from Station 15 (Pingtan) to Station
19 (Yunwo) along the East China Sea coast, where LA-HadISST indicated a stronger
warming than in the LH, is shown in Fig. 2c. The local data indicate markedly lower
temperatures, which may mainly be because of coastal upwelling (the effect of upwelling will
be discussed in the Section 5), but also other local effects, including local tidal mixing, ocean
fronts, sea water vertical mixing, and fresh water discharge, etc., but also a weaker trend (0.18
℃ per decade) than in the LA-HadISST (0.35 ℃ per decade).
**Table 3**. Statistics of the time series of the localized SST-analysis (LA-HadISST) data series at the 26
station, as well as the differences (Diff) between statistics of the LH series given in Table 1. The correlation
coefficients between LH and LA-HadISST are also calculated (the 90% confidence level is 0.22, without
considering serial correlation). Red numbers indicate that the correlation coefficients do not conflict with
the null hypothesis of no correlation.

| Station | Mean | Diff | Std deviation | Diff | Trend | Diff | Corr |
|---------|------|------|---------------|------|-------|------|------|

| No. | LA-HadISST | | LA-HadISST | | (℃/10yrs) | | |
|---|---|---|---|---|---|---|---|
| 1 | 12.80 | -1.32 | 0.43 | -0.06 | -0.02 | 0.25 | **0.20** |
| 2 | 12.93 | -0.72 | 0.37 | 0.21 | 0.02 | 0.24 | 0.31 |
| 3 | 13.45 | -1.76 | 0.46 | 0.38 | 0.13 | 0.16 | 0.73 |
| 4 | 13.86 | -2.30 | 0.51 | 0.07 | 0.15 | 0.07 | 0.67 |
| 5 | 13.71 | -0.24 | 0.54 | 0.28 | 0.11 | 0.11 | 0.66 |
| 6 | 13.92 | -1.12 | 0.57 | 0.01 | 0.14 | 0.03 | 0.69 |
| 7 | 14.87 | -2.58 | 0.58 | 0.01 | 0.19 | -0.05 | 0.70 |
| 8 | 14.51 | 0.01 | 0.54 | 0.10 | 0.14 | 0.03 | 0.77 |
| 9 | 14.51 | -0.60 | 0.54 | 0.08 | 0.14 | 0.08 | 0.66 |
| 10 | 16.05 | -1.07 | 0.47 | 0.10 | 0.21 | 0.00 | 0.71 |
| 11 | 19.70 | -2.00 | 0.57 | 0.08 | 0.12 | 0.14 | 0.63 |
| 12 | 20.66 | -2.65 | 0.59 | 0.05 | 0.27 | -0.03 | 0.67 |
| 13 | 20.66 | -2.12 | 0.59 | -0.03 | 0.27 | -0.10 | 0.64 |
| 14 | 22.47 | -2.30 | 0.70 | 0.01 | 0.35 | -0.14 | 0.73 |
| 15 | 23.43 | -3.00 | 0.75 | -0.14 | 0.34 | -0.15 | 0.65 |
| 16 | 23.43 | -1.45 | 0.77 | -0.25 | 0.40 | -0.23 | 0.75 |
| 17 | 22.03 | -2.41 | 0.77 | -0.26 | 0.40 | -0.21 | 0.78 |
| 18 | 24.46 | -3.26 | 0.59 | -0.14 | 0.30 | -0.17 | 0.59 |
| 19 | 24.46 | -3.08 | 0.59 | -0.15 | 0.30 | -0.17 | 0.66 |
| 20 | 25.44 | -2.82 | 0.46 | -0.02 | 0.20 | -0.05 | 0.83 |
| 21 | 25.66 | -1.78 | 0.51 | -0.01 | 0.07 | 0.11 | 0.56 |
| 22 | 25.11 | -1.47 | 0.31 | 0.24 | 0.07 | 0.11 | 0.53 |
| 23 | 25.11 | 0.71 | 0.31 | 0.13 | 0.07 | 0.10 | 0.41 |
| 24 | 25.65 | -1.02 | 0.40 | 0.09 | 0.19 | -0.03 | 0.55 |
| 25 | 25.65 | -0.47 | 0.40 | 0.09 | 0.19 | -0.03 | 0.57 |

| 26 | 25.93 | 0.09 | 0.43 | 0.00 | 0.22 | -0.04 | 0.64 |


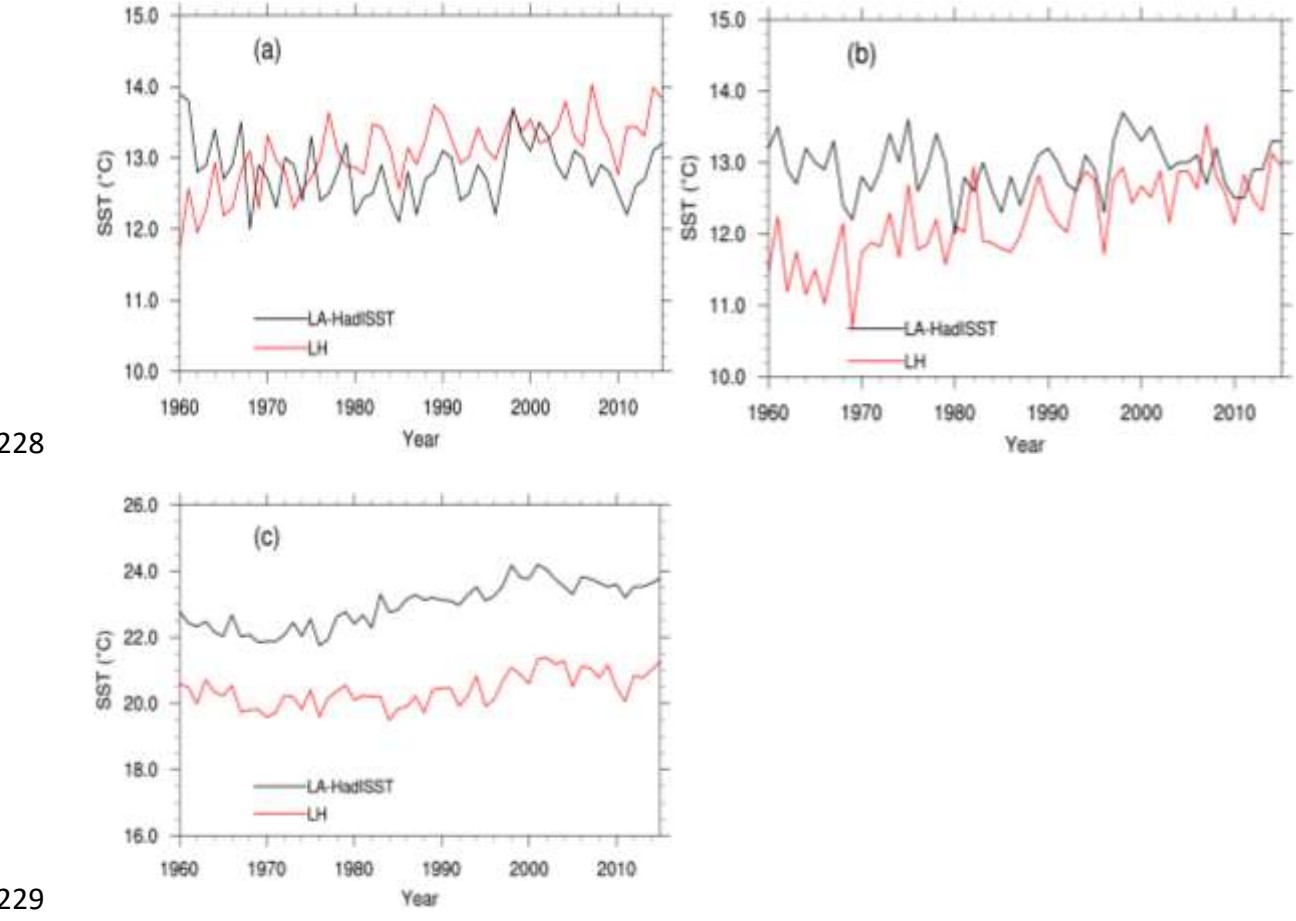

**Figure 2**. The annual mean SST series of LA-HadISST (black line) and LH (red line) from Station 1 (Zhimaowan) **(a)** and Station 2 (Qinhuangdao) **(b)**; The average annual mean SST series of LA-HadISST (black line) and LH (red line) from Station 15 (Pingtan) to Station 19 (Yunwo) **(c)**.

The first two EOFs of the LH and the LA data set have similar patterns, namely a uniform sign along the entire coast in EOF1, with similar intensities, and a north-south dipole (Bohai Sea and Yellow Sea vs. East and south China Sea) in EOF2, with a sign change at Station 11 (Shipu) (Fig.3a and 3b). The two patterns of LH explain less variance, namely 82.9% of the total variance, than the LA-HadISST EOFs, which go with 92.9%. This may be related to the larger spatial variability in local data compared to gridded data. In EOF1, again the Station 1 (Zhimaowan) and Station 2 (Qinhuangdao) in the Bohai Sea contribute less in LA-HadISST, whereas the Station 15 (Pingtan) to Station 19 (Yunwo) contribute more to the overall warming than in LA-HadISST than in LH.

The time coefficients (PCs) are broadly similar, even if the correlations are not very strong: only 0.84 and 0.42 (Fig.3c and 3d). A general warming is associated with EOF1 and mostly stationary inter-annual variability with EOF2. Again, the sequence of maxima and minima is

qualitatively similar, but PC2 of LA-HadISST exhibits a break point at about 1980 –
interestingly the time when satellites became routinely available of the global analyses. These
data improve SST sampling, especially in the Southern Ocean and coastal areas (Smith et al.,
2008; Lima and Wethey 2012). Before 1980, PC2 of LH and LA-HadISST differed by about
0.2 (Fig. 3d; this corresponds to a mean difference of 0.04K at the southern stations from
Station 11 to Station 26 during that time, and a mean difference 0.04K at the northern stations
from Station 1 to Station 10 (Fig. 3b)).
To further study the differences in trends, EOFs were calculated from the difference time
series, that is, LH anomalies minus LA-HadISST anomalies at the 26 sites (Fig. 4). The first
two EOFs stand for 31.2% and 27.6% of the variance. These numbers are not very different,
and it their closeness may be indicative that the EOFs are degenerate (von Storch and Zwiers
1999). These EOFs describe covariations of the differences along long stretches of the coast;
in case of EOF1, this is the case for all stations at the southern Station 11 (Shipu), i.e., in the
East and South China Sea (Fig.4a). In EOF2 it is all stations at the southern Station 13
(Kanmen), mostly in the Yellow Sea and Bohai Sea (Fig. 4b). PC1 seems to describe a change
point at about 1980, whereas PC2 describes a slight upward trend: The differences tend to be
larger in earlier years and are almost nil in the end of the considered time interval. That is, in
recent years, there are little differences between LA-HadISST and LH, which is not surprising
giving the better observational and reporting practice.
That in early years inhomogeneities impacted the quality of SST analyses is also not
surprising, but it is valuable to learn when these inhomogeneities took place, and which time
periods in the analyses should be taken with some reservation. Of course, this assertion
depends on the assumption that the homogenization of the local data did remove all change
points and other inhomogeneities.

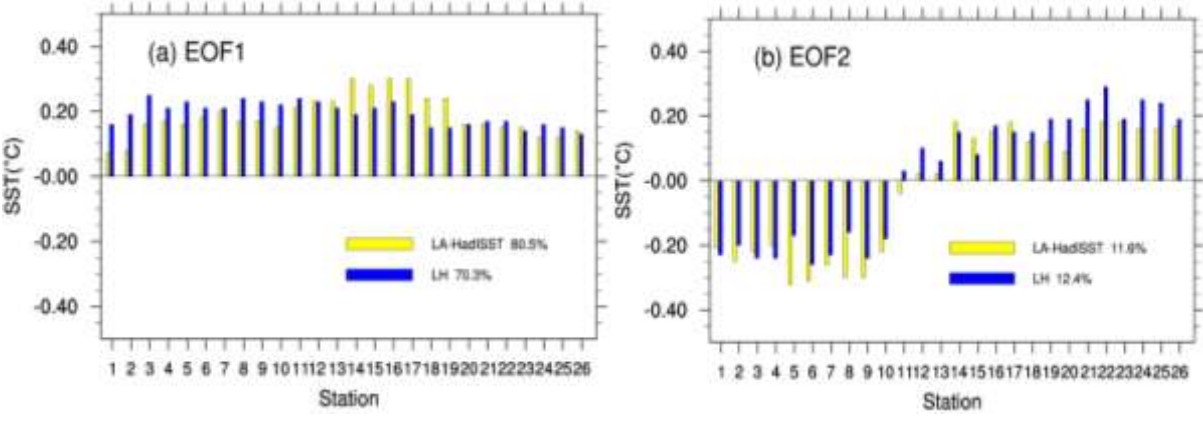


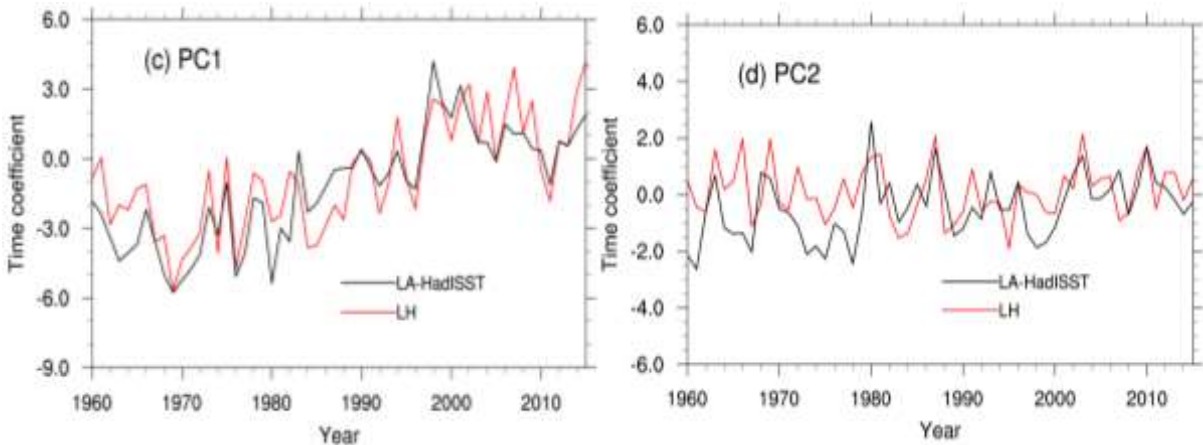


**Figure 3**. Comparison of the EOF1 and EOF2 derived from the LH data set of local SST at 26 sites (blue
bars; red lines), and derived from the localized analysis data LA-HadISST (yellow bars; black lines).
Top: EOF spatial patterns, bottom: principal components (time coefficients).


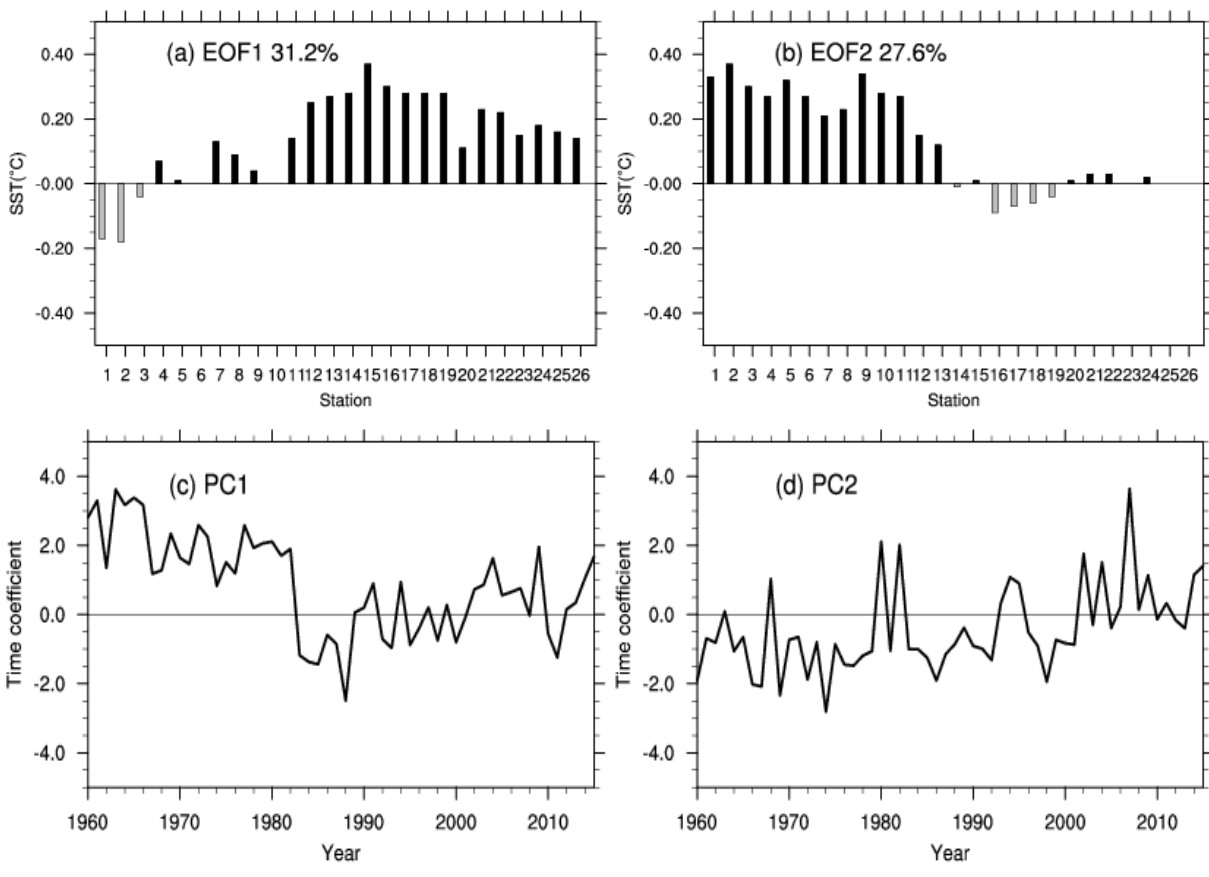


**Figure 4**. First two EOFs of the difference time series LH-LA-HadISST. Top: EOF spatial patterns, bottom:
principal components (time coefficients).


## 4.2 Comparing with COBE SST

In this subsection, we consider the localized SST derived from the LA-COBE SST data set during 1960–2015. Again, the LA-COBE SST is in almost all sites higher than the local data, namely at 21 out of 26 sites. The differences are up to 3K, and again mostly along the East China Sea coast from Station 11 (Shipu) to Station 20 (Zhelang) (see Table SOM-1 in the Supplementary Online Material (SOM)). The local correlations are relatively high, namely between 0.55 and 0.85.

The EOFs derived from the LA-COBE SST, with the same grid resolution of $1^{o}$ and the same time window 1960–2015 as LA-HadISST, exhibits broadly the same pattern in space and time as the EOFs of the LH data. Also, the explained variances are close (Figure SOM-2). The northern stations contribute more to the overall warming represented by EOF1, whereas the stations along the South and East China Sea contribute less. Again, the two northernmost Station 1 (Zhimaowan) and Staation 2 (Qinhuangdao) exhibit some systematic differences, both in EOF1 and EOF2. The PCs share correlations of 0.80 for EOF1 and 0.50 for EOF2. COBE SST does not capture the recovery of the dip in warming since about 2000, as LH and HadISST did, while EOF2 reveals some warming in the final years. During the 1960s some differences prevail.

Fig.5 shows the EOFs of the difference time series between LH anomalies and LA-COBE SST anomalies. The first EOF dominates, with 49.8%, whereas the second one represents a share of 17.5%. The first EOF points to several inhomogeneities, with two prolonged intervals during which LH is higher than LA-COBE SST (i.e., 1960–1978, and 1995–2005), and a strong drop-down do negative PC-values after about the year of 2005. PC2, on the other hand, appears as mostly stationary, except for a suspiciously negative episode in the early 1960s.

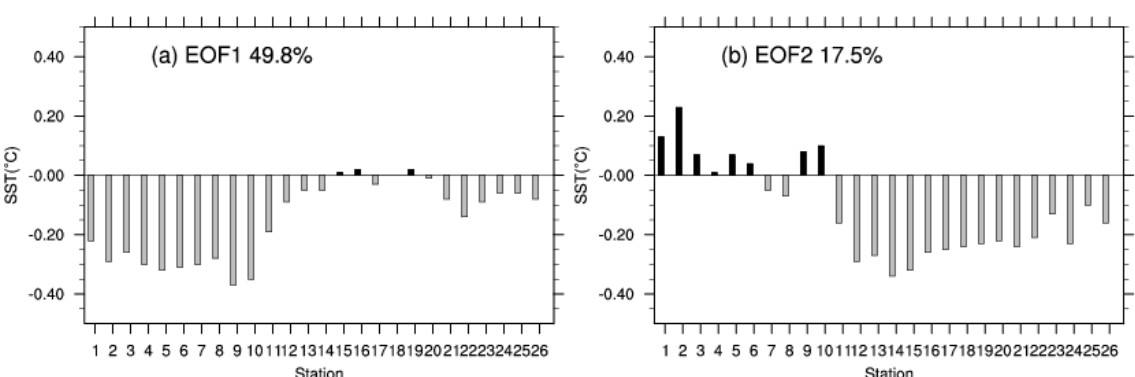

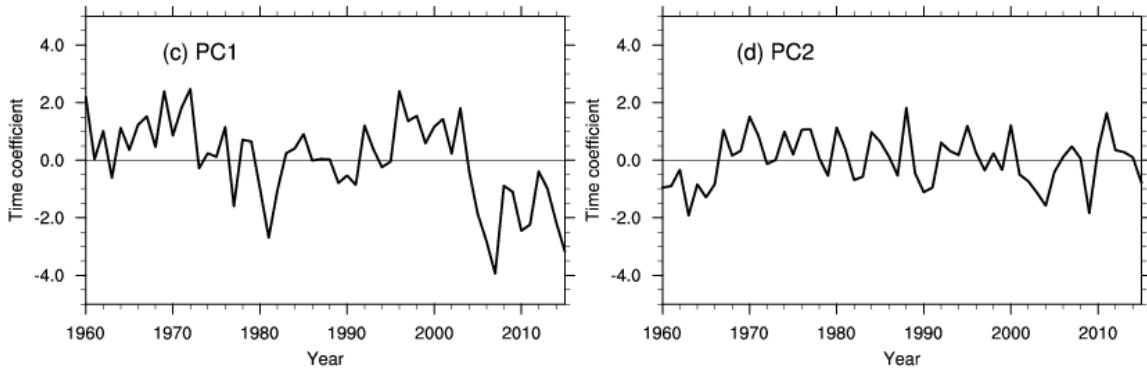

304

**Figure 5**. EOF analysis of the differences LH-LA-COBE: Top: EOF spatial patterns (EOFs), bottom: principal components (time coefficients).

## 4.3 Comparing with ERSST

ERSST presents SST on a coarser grid compared to the two cases before. Again, the temperatures given by ERSST, as was the case with the other two analyses, are higher than the temperatures recorded at the local sites along the coast (see Table SOM-2). The differences are up to 4K, and the largest differences are found in the East China Sea from Station 11 (Shipu) to Station 20 (Zhelang). That the differences are in this case even larger than in the other LA cases may be related to the $2^o$ coarse resolution of ERSST.

The variability according to ERSST is quite similar to that of LH, at least in terms of EOFs (see Figure SOM-3). The correlation of the PC1's is 0.83, and of PC2's to 0.60. LA-HadISST got 0.84 and 0.42, LA-COBE SST got 0.80 and 0.50. The local correlations vary between 0.37 and 0.82. Again EOF1 stands for an overall warming and EOF2 to interannual variability with hardly a trend. The relative contributions of the two EOFs compare well to the LH-EOFs. In detail, the northernmost stations appear stronger in EOF1 of LA-ERSST than in that of LH, whereas the northern sites are underrepresented, and the southern over-represented in EOF2.

The EOFs of the differences between LH anomalies and LA-ERSST anomalies are shown in Fig. 6. They differ strongly from those found for LA-COBE SST and LA-HadISST. The first EOF differences resemble the first EOFs of LH and LA-ERSST (not shown; see Fig. SOM-3) – the long-term trend in LA-ERSST is smaller than in the local data – everywhere. The second EOF is again a dipole pattern, with the Bohai Sea and the Yellow Sea on the one side, and the East China Sea and South China Sea on the other. The time series of PC2 fluctuates around zero without prominent long-term trend.

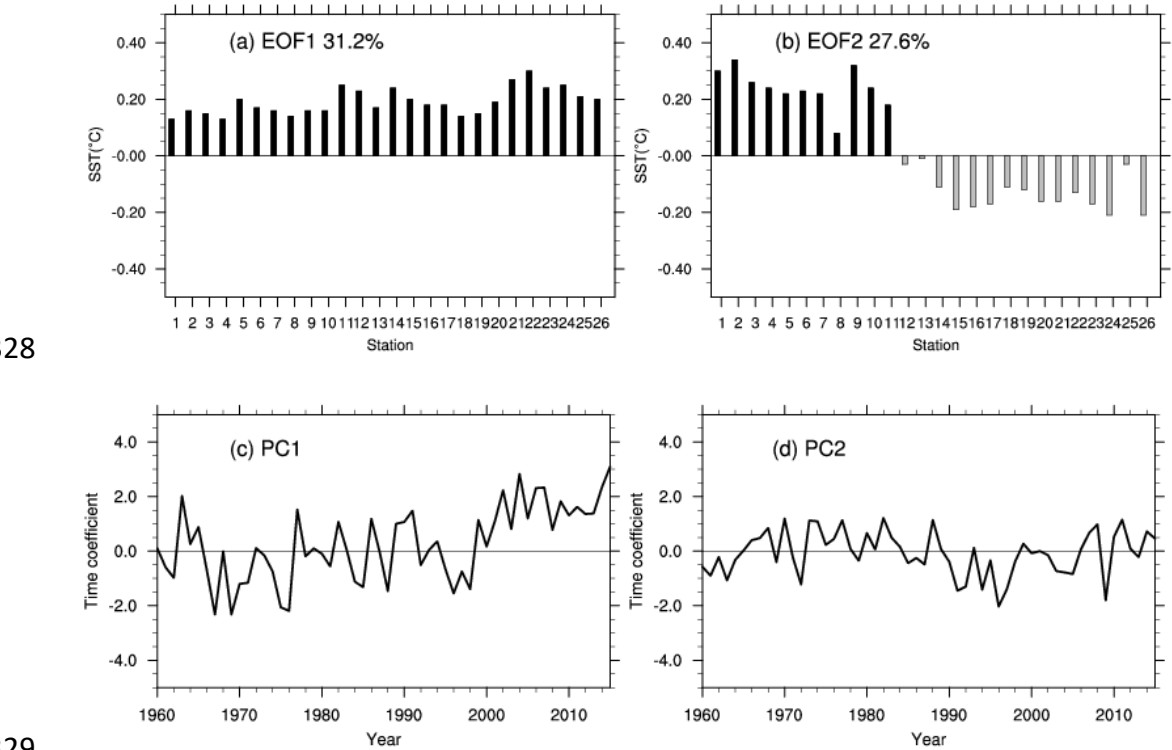

328

329

**Figure 6**. EOF analysis of the differences LH-LA-ERSST: Top: EOF spatial patterns (EOFs), bottom:
principal components (time coefficients).

## 5 Discussion and conclusion

We have mainly examined three global gridded analysis SST data sets in the Chinese coastal
waters. For doing so, we have compared a number of statistical properties for 26 coastal
hydrological locations as given by the analyses and by a newly digitized and homogenized
data set (Li et al., 2018). For demonstrating the utility of the local data set, we have compared
the local SST series (named LH) with independent local homogenized SAT data from nearby
meteorological stations. The variations of the two series are fully consistent. Another
argument points to the quality of the LA data set is that the differences between LH and the
three LAs (localized data from the different global analyses: HadISST1, COBE SST, ERSST)
considered are not uniform (except for the time mean); instead the LAs deviate in different
ways from LH. If this would not be the case, one could be tempted to argue that the
differences are manifestations of inefficiencies of the LH data set. This is not the case.

In this study, we found that all of these globally gridded datasets exhibit surface temperatures
usually higher than the LH data, especially at the East China Sea. This difference may be
caused by two factors. In the China Seas, most of the coastal upwelling currents occur at the
East China Sea and the northern South China Sea, other small upwelling currents at the tops

of the Liaodong Peninsula and Shandong Peninsula (Figure 1) (Yan 1991). The consensus of
previous studies is that coastal upwelling currents results in cooling SST at these coastal areas
(Xie et al, 2003; Guan et al., 2009; Su et al., 2012). In our study, we find that the in-situ
shoreline SSTs at the upwelling areas (e.g. Station 4 (Laohutan), Sation 11 (Shidao) and
Station 18 (Dongshan)) are always colder than global gridded SST data, with the value of
below -1K (Table 2, Table 3 and Table SOM1).
We hypothesize that these negative differences are connect by coastal upwelling currents. To
test this hypothesis, we examine the output of a numerical simulation of the currents in the
South China Sea with a grid resolution of 0.04 °. The model is embedded in an almost global
model with 1 ° grid resolution (Tang et al., 2018). The model used here is the Hybrid
Coordinate Ocean Model (HYCOM) that is exposed to periodic climatological atmospheric
forcing, with a fixed annual cycle but no weather disturbances. The atmospheric forcing
comes from the International Comprehensive Ocean-Atmosphere Data set (ICOADS). We
extract simulated SSTs at three different distances (i.e., near the station, 50 km, and 100 km
from each coastal hydrological station in South China Sea). Fig.7 shows that most shoreline
SSTs are lower than ambient offshore SSTs, especially SSTs at 100km from shoreline.
However, the Stations 22 (Beihai) and Station 23 (Weizhou) are not affected by coastal
upwelling, and consistently, there are no notable differences among SSTs at three different
distances from the two stations. The result reflects that the homogenized SST data set for
shoreline stations catches this relative cooling water effect of the regional upwelling currents.
On the other hand, the global gridded SST datasets point to higher temperatures which may be
caused by their coarse resolution. The differences are largest in the case of the coarsest
analysis (ERSST), but weakest in the OISST v2 analysis with a resolution of 0.25 ° (Fig. 8; see
below) (Note that the difference of LH minus LA-OISST is restricted to the warmer episode
1982–2015). Meanwhile, the lack of near-shore observations when compiling near-shore box
averages in coastal areas may also cause these differences (Wang et al., 2018). Besides, there
still some other local mechanisms with smaller scale can cause cooling water in the China
Seas, such as China coastal current (CCC) (Belkin and Lee, 2014) and ocean fronts (Zhao,
1987; Hickox et al., 2000). In them, the shallow water shelf front and estuarine plume front
are two major fronts in the Bohai Sea and the Yellow Sea in summer. Coastal current front,
upwelling front as well as strong westerly boundary current usually appears in the East China
Sea and the South China Sea which may also be related to coastal upwelling currents.
In summary, our main results are as follows:
– The mean SST in LH at many sites is considerably lower than that in the LA-data sets.
We suggest that this is related to local oceanic effects, such as coastal upwelling. The
LA-datasets cannot catch this cooling effect of the regional upwelling currents well.
On the other hand, the global gridded SST datasets point to higher temperatures which
may be caused by their coarse resolution when averaging in the LA data sets. However,
systematic differences would not be expected to influence strongly the overall
variability and trends.

– The first EOF in all data sets stands for a general warming, and the second for
interannual variability. This is not only so in the local LH-data but also in all globally
gridded-based LA-datasets.

– In the years following the introduction of satellites in monitoring SST, since about
1980, the different global analyses converge, and the differences to the local data set
become smaller. In support of this, the comparison with the high resolution analysis
OISST v2 for the post-satellite period 1982–2015 reveals few differences (not shown,
see Fig. SOM-4).

– In the years before 1980, some noteworthy differences are found. The differences
between the LH-data anomalies and the LA-data anomalies are non-uniform across the
different LA data sets. For instance, for ERSST the long-term trends differ, in case of
COBE SST several jumps emerge, and in case of HadISST, a jump is found at the time
of the advent of the routine satellite data, but also a trend in PC2 of the differences.

Thus, our overall conclusion is that the global gridded SST datasets correctly describe the
main features of variabilities and trends in regional waters, but that significant improvements
in the regional analyses may be gained when quality controlled homogenized data are
incorporated. In particular for the time prior to the usage of remote sensing by satellites, and
in regions where observational efforts have been limited, such efforts are valuable
contributions to climate variability and change studies. Our example should also be an
encouragement for national climate services to revisit regional data, and to invest into the
elimination of inconsistencies caused by inhomogeneities. There are several projects or
researches dedicated quality control and homogenization of *in situ* data (Kuglitsch et al., 2012;
Hausfather et al., 2016; Minola et al., 2016). It is useful to keep some high-quality data
separate from that available for analyses, for validation activities such as our work and others'
work (Hausfather et al., 2017).

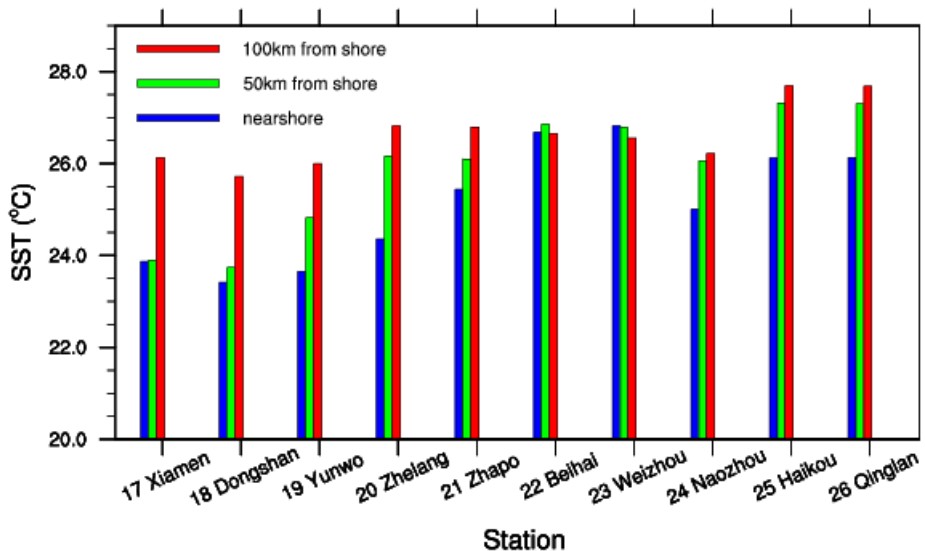


**Figure 7**. Simulated SSTs at different distances from each coastal hydrological station in the South China
Sea.

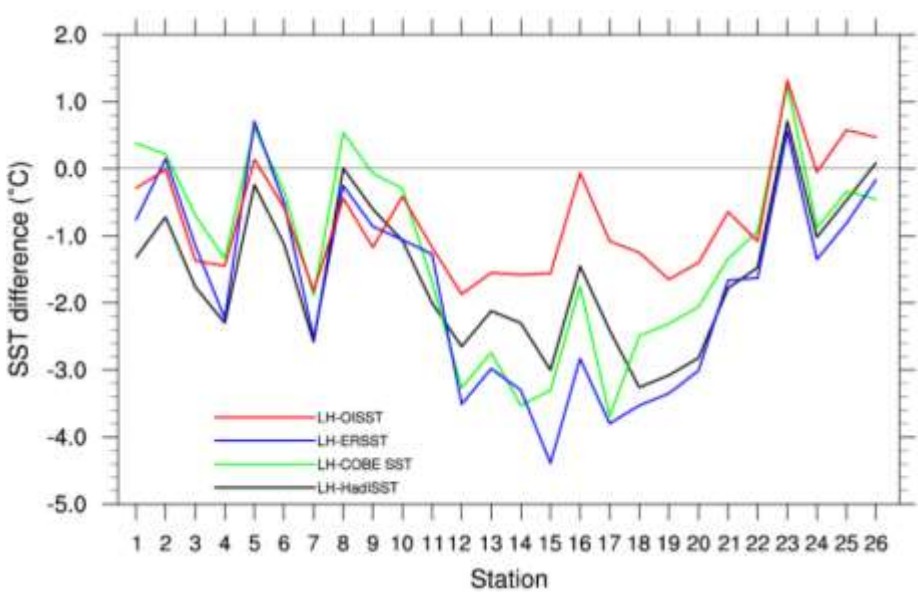


**Figure 8.** The mean SST differences at the 26 locations between LH and LA-OISST (1982-2015; red
line), LH and LA-ERSST (1960-2015; blue line), LH and LA-COBE SST (1960-2015; green line) and
LH and LA-HadISST (1960-2015; black line)

*Data availability.* All the four gridded SST analyses used in this study are publicly available
and can be downloaded freely from the websites shown in Table 1. The observational in situ
SST data from the coastal stations and the coordinates of coastal stations can be obtained from
the National Marine Science Data Center, National Science & Technology Resource Sharing
Service Platform of China (http://mds.nmdis.org.cn). However, the observational in situ SST

data from only 9 coastal stations are publicly available. SST data from the rest stations can be obtained after an application to the website.

*Acknowledgments*. The work is funded by the program of National Natural Science Foundation of China (No. 41376014; No. 41706020), the National Key Research and Development Program of China (No.2018YFA0605603; No. 2017YFC1404700) and also supported by the Hamburg University's Cluster of Excellence CliSAP in Germany, Shengquan Tang's work is funded by the Chinese Scholarship Council.

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

**Appendix A: Consistency of homogenized SST data set with homogenized SAT data set**

We examine if the SST data is consistent with other local homogenized data, specifically with time series of surface air temperature (SAT) at various locations along the Chinese coast. This data set contains data from many sites. For each of the SST measuring sites, there is at least one SAT stations within 100 km distance. We form 26 pairs of located SST/SAT data. SST and SAT data directly are not compares pairwise, but in terms of the patterns and coefficient time series (PCs) of their empirical orthogonal functions (EOFs).

The first EOFs of SST and *in situ* SAT describe an overall warming, with a slight tendency of stronger warming in terms of both SST and SAT in the northerly Bohai and Yellow Sea (Fig. A1a). This pattern is dominant, representing 70.3% and 76.7% of the total interannual variance. The warming is mostly continuous from about 1970 until 2010 (Fig. A1c). The similarity of the principal components – expressed by 0.97 in terms of the correlation coefficient – is striking (Fig. A1c). The second EOFs represent considerably less variance – namely about 11.6% (Fig. A1b). They describe a North-South contrast, and stationary PCs, varying around 0 without prolonged positive or negative excursions (Fig. A1d). Also the PCs of the second PCs of SST and SAT show a remarkably parallel development – with a high correlation of 0.86 (Figs.A1d).

When this exercise is repeated with CRU TS 3.24.01 instead of the *in situ* SAT series, we find similar consistency (see Fig. SOM-1). The PCs of SAT-CRU also show high correlations of 0.94 and 0.83 with the *in situ* SST (see Fig. SOM-1).

We conclude that the two data sets are consistent; the first EOFs describe the warming of the recent decades of years; the second EOFs describe interannual variability, and may be influenced by ENSO and other patterns of natural variability. We furthermore conclude that the new description of SST variability and trends at the 26 sites along the Chinese coast presents a reliable account of the past since 1960 – and thus may serve as a benchmark for assessing global analyses of SST datasets.

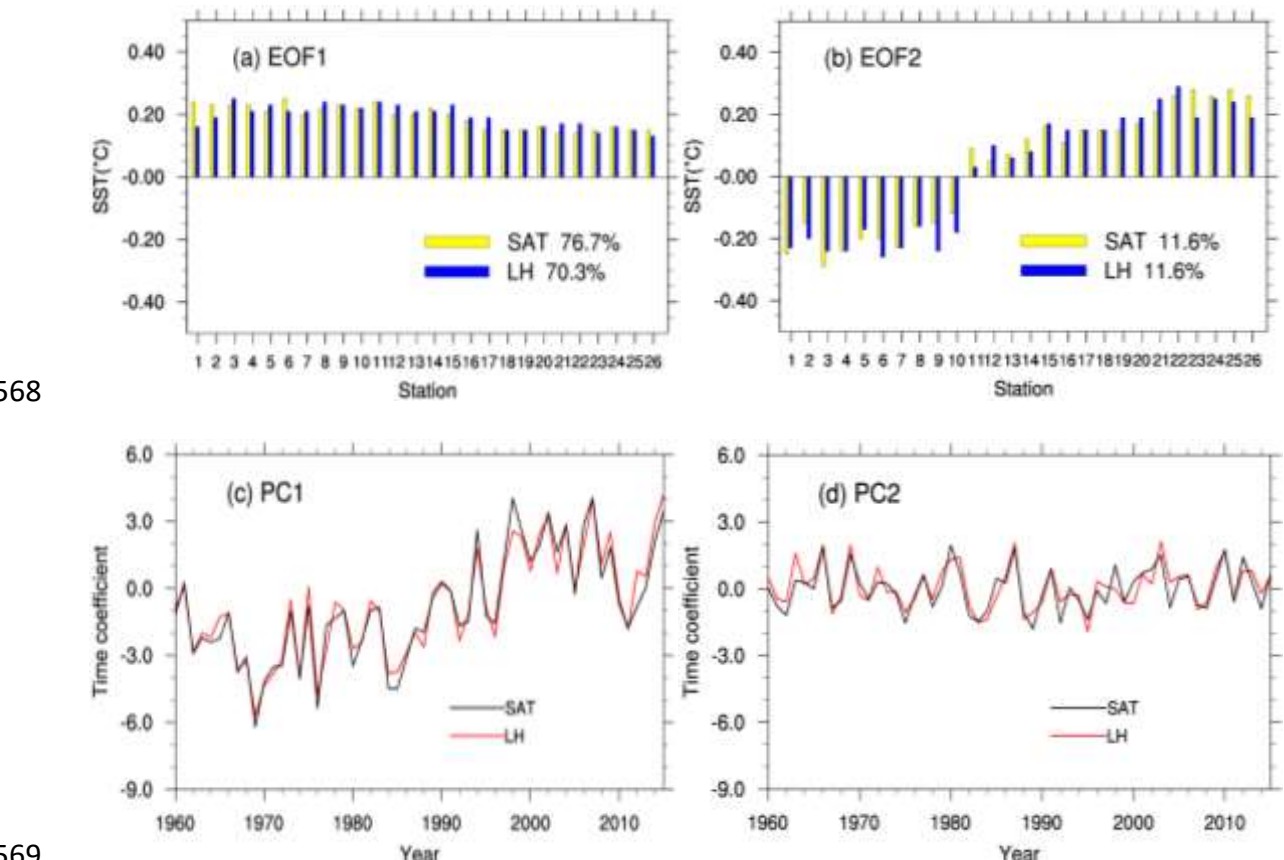


**Fig. A1.** Comparison of the EOF1 and EOF2 derived from the LH data set of local SST at 26 sites (blue
bars; red lines), and derived from the SAT at the same sites (yellow bars; black lines).
Top: EOF spatial patterns, bottom: principal components (time coefficients).


        **Appendix B: Supplementary Online Material (SOM)**


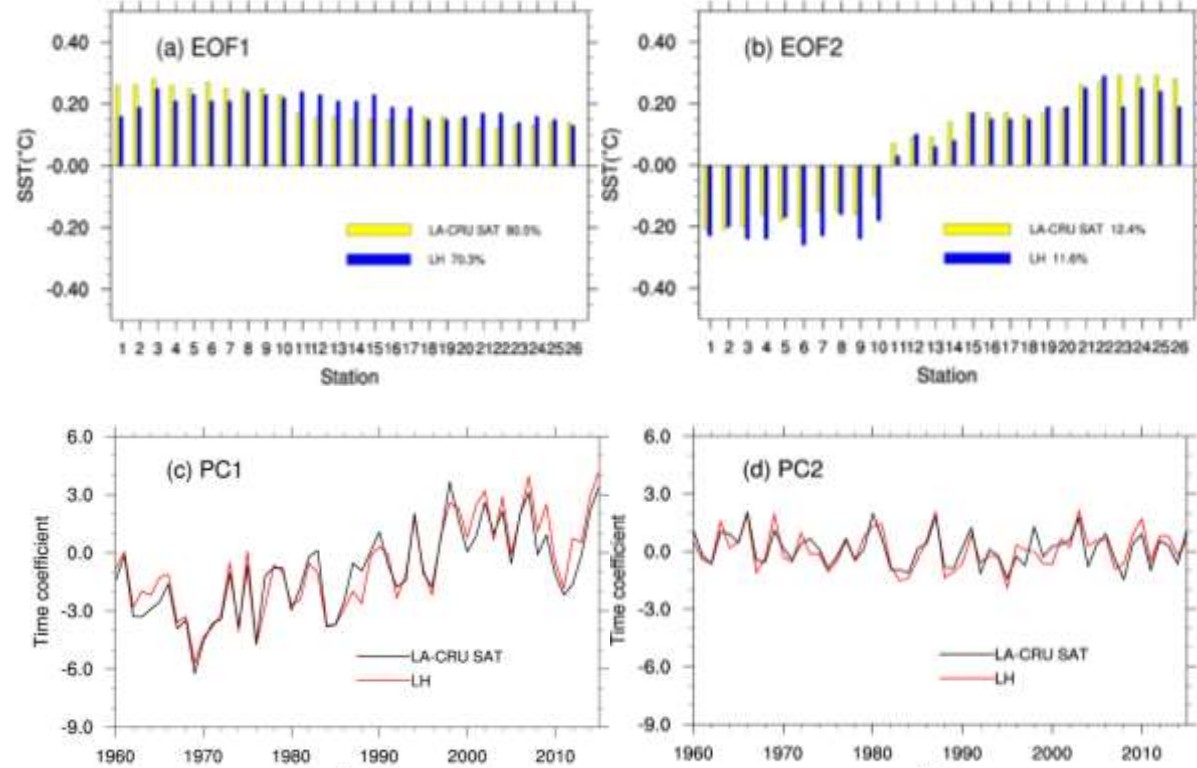



**Fig. SOM-1.** Comparison of the EOF1 and EOF2 derived from the LH data set of local SST at 26 sites

(blue bars; red lines), and derived from the CRU SAT at the same sites (yellow bars; black lines).

Top: EOF spatial patterns, bottom: principal components (time coefficients).


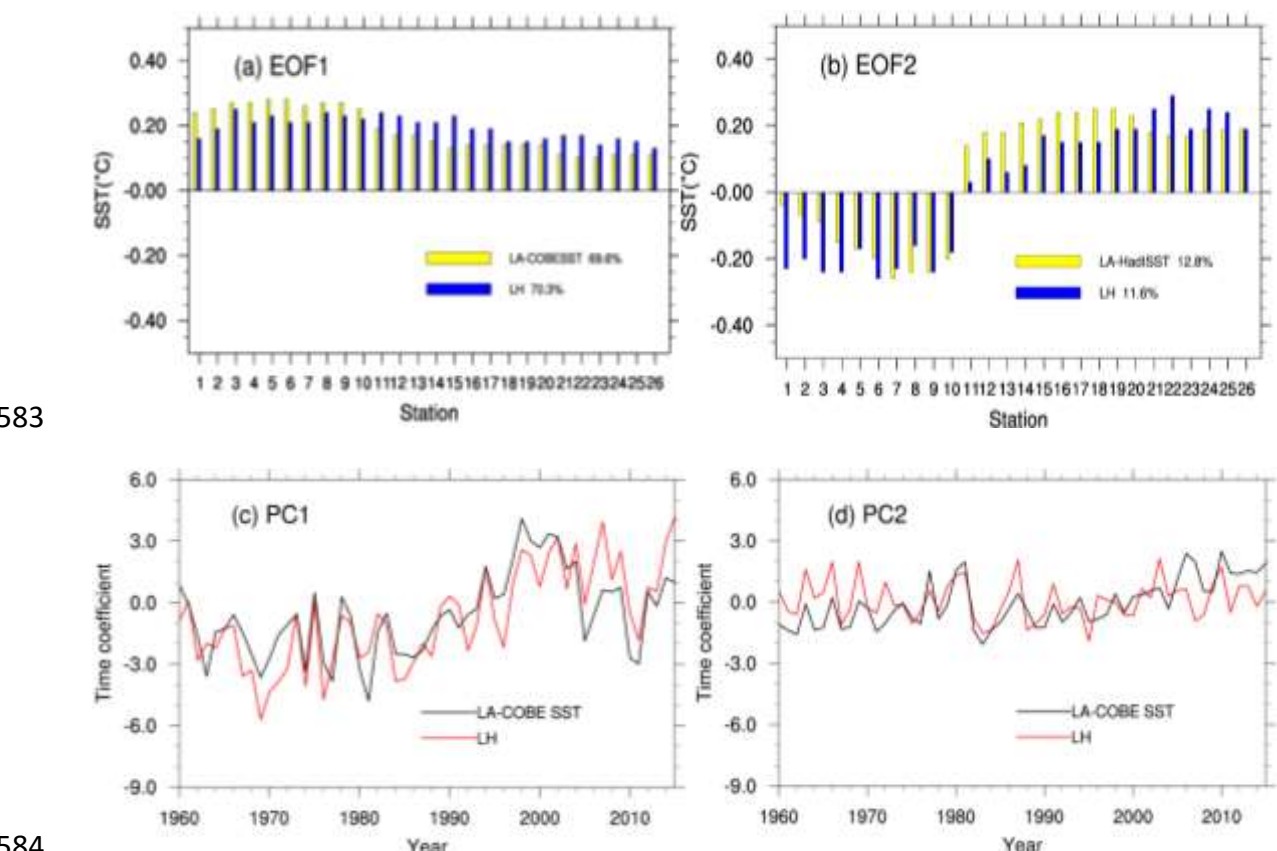



**Fig. SOM-2.** Comparison of the EOF1 and EOF2 derived from the LH data set of local SST at 26 sites
(blue bars; red lines), and derived from the localized analysis data LA-COBE SST (yellow bars; black
lines). Top: EOF spatial patterns, bottom: principal components (time coefficients).

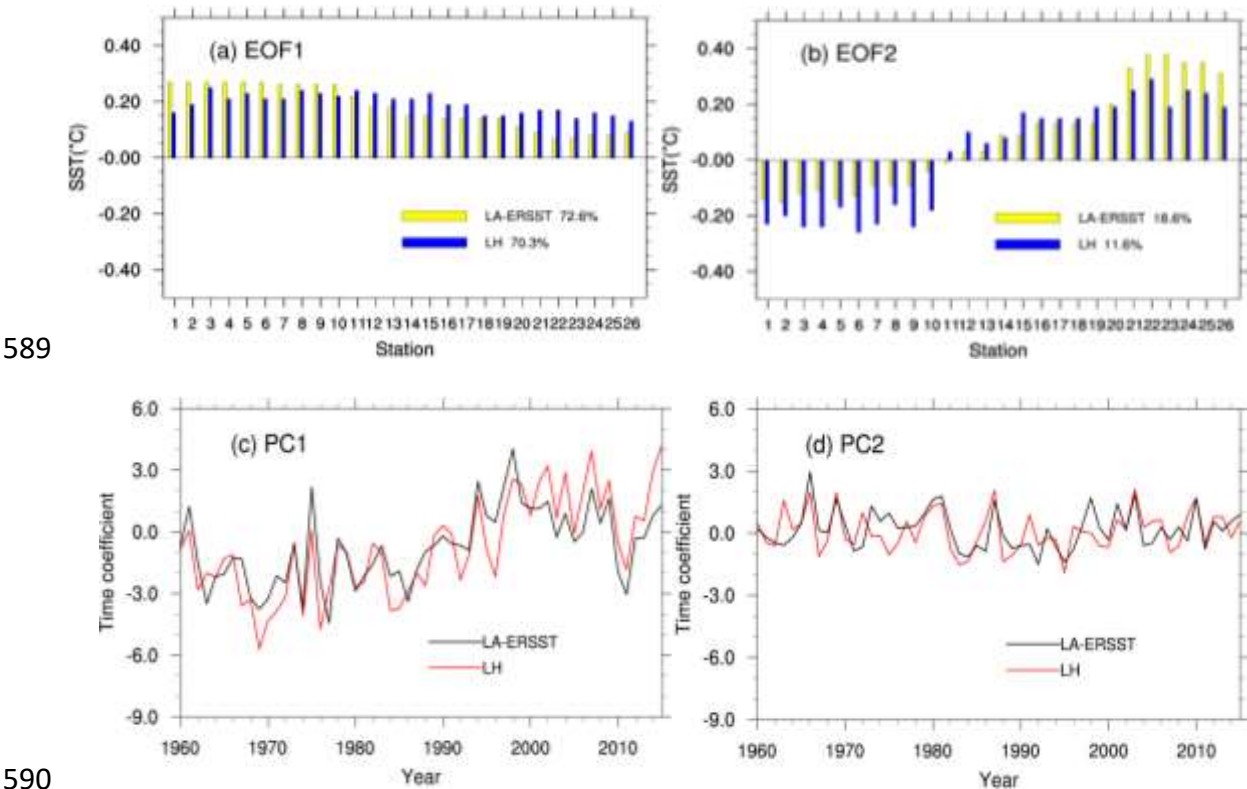




**Fig. SOM-3.** Comparison of the EOF1 and EOF2 derived from the LH data set of local SST at 26 sites
(blue bars; red lines), and derived from the localized analysis data LA-ERSST (yellow bars; black lines).
Top: EOF spatial patterns, bottom: principal components (time coefficients).

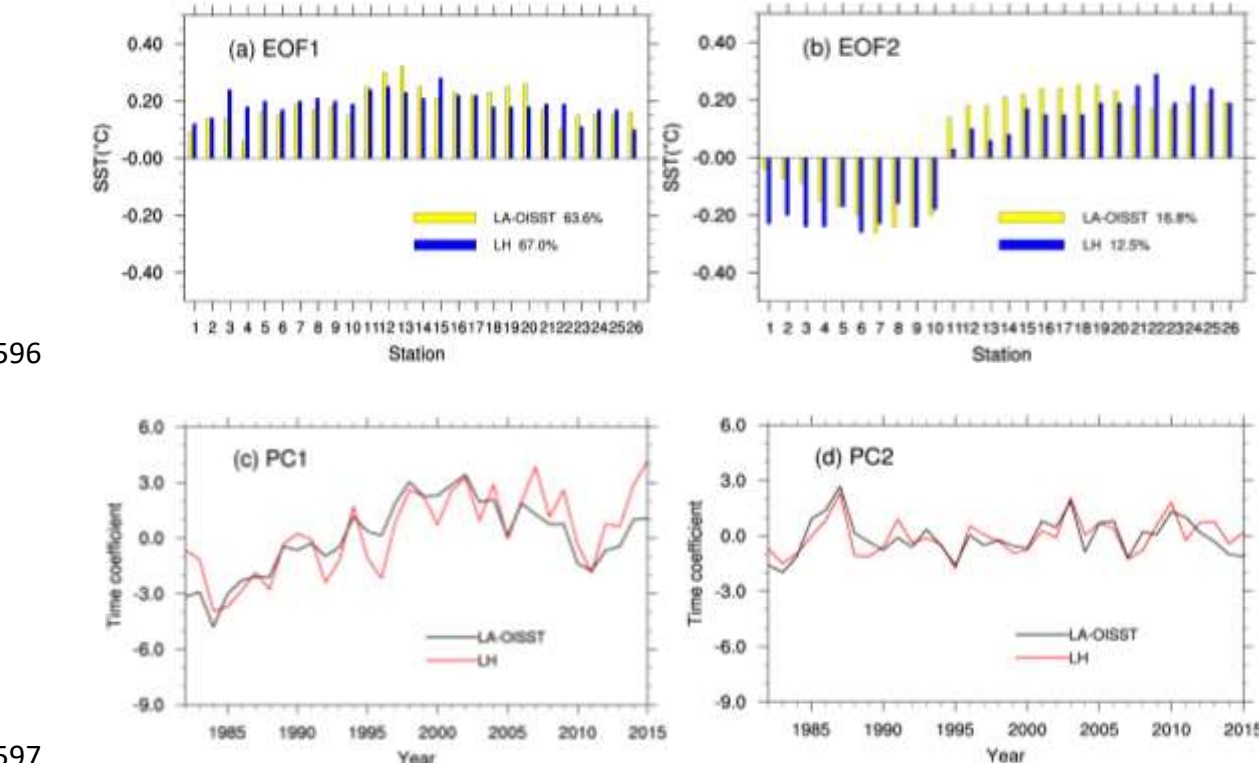



**Fig. SOM-4.** Comparison of the EOF1 and EOF2 derived from the LH data set of local SST at 26 sites
(blue bars; red lines), and derived from the localized analysis data LA-OISST (yellow bars; black lines).
Top: EOF spatial patterns, bottom: principal components (time coefficients).


Table SOM-1. Statistics of the time series of the localized SST-analysis (LA-COBE SST) data series at the 26 station, as well as the differences (Diff) between the pairs of time series. The correlation coefficients between LH and LA-COBE SST are also calculated (the 90% confidence level is 0.22, without considering serial correlation). Red numbers indicate that the correlation coefficients do not exceed the 90% confidence level.

| No | Mean LA-COBE SST | Diff | Std-dev LA-COBE SST | Diff | Trend (°C/10yrs) | Diff | Corr |
|----|------|------|------|------|------|------|------|
| 1  | 11.13 | 0.38 | 0.52 | 0.01 | 0.17 | 0.00 | 0.60 |
| 2  | 11.99 | 0.22 | 0.54 | 0.04 | 0.16 | 0.10 | 0.56 |
| 3  | 12.23 | -0.69 | 0.56 | 0.14 | 0.14 | 0.15 | 0.74 |
| 4  | 12.70 | -1.34 | 0.59 | 0.00 | 0.10 | 0.11 | 0.59 |
| 5  | 12.75 | 0.61 | 0.60 | -0.01 | 0.10 | 0.12 | 0.64 |
| 6  | 12.98 | -0.33 | 0.61 | -0.03 | 0.07 | 0.10 | 0.66 |
| 7  | 13.98 | -1.89 | 0.61 | -0.02 | 0.01 | 0.13 | 0.68 |
| 8  | 13.83 | 0.54 | 0.62 | 0.03 | 0.04 | 0.13 | 0.72 |
| 9  | 13.83 | -0.07 | 0.62 | 0.01 | 0.03 | 0.19 | 0.55 |
| 10 | 15.14 | -0.29 | 0.57 | 0.00 | 0.03 | 0.18 | 0.55 |
| 11 | 19.09 | -1.68 | 0.45 | 0.20 | 0.18 | 0.08 | 0.77 |
| 12 | 20.94 | -3.27 | 0.43 | 0.22 | 0.19 | 0.05 | 0.81 |
| 13 | 20.94 | -2.74 | 0.43 | 0.13 | 0.19 | -0.02 | 0.78 |
| 14 | 23.25 | -3.53 | 0.38 | 0.22 | 0.20 | 0.01 | 0.82 |
| 15 | 23.29 | -3.30 | 0.41 | 0.11 | 0.20 | -0.01 | 0.79 |
| 16 | 23.29 | -1.75 | 0.41 | 0.10 | 0.20 | -0.03 | 0.85 |
| 17 | 22.90 | -3.69 | 0.40 | 0.14 | 0.19 | 0.00 | 0.78 |
| 18 | 23.33 | -2.49 | 0.41 | 0.04 | 0.21 | -0.08 | 0.68 |
| 19 | 23.33 | -2.31 | 0.41 | 0.02 | 0.21 | -0.08 | 0.77 |
| 20 | 24.49 | -2.06 | 0.40 | 0.04 | 0.18 | -0.03 | 0.81 |
| 21 | 24.95 | -1.34 | 0.33 | 0.17 | 0.11 | 0.07 | 0.80 |
| 22 | 24.53 | -0.93 | 0.34 | 0.21 | 0.10 | 0.08 | 0.78 |
| 23 | 24.53 | 1.26 | 0.34 | 0.09 | 0.10 | 0.07 | 0.73 |
| 24 | 25.34 | -0.88 | 0.35 | 0.14 | 0.12 | 0.04 | 0.77 |
| 25 | 25.34 | -0.34 | 0.35 | 0.13 | 0.12 | 0.04 | 0.85 |
| 26 | 26.25 | -0.45 | 0.36 | 0.08 | 0.13 | 0.05 | 0.68 |

Table SOM-2 Statistics of the time series of the localized SST-analysis (LA-ERSST) data series at the 26 station, as well as the differences (Diff) between the pairs of time series. The correlation coefficients between LH and LA-ERISST are also calculated (the 90% confidence level is 0.22, without considering serial correlation). Red numbers indicate that the correlation coefficients do not exceed the 90% confidence level.

| No | Mean LA-ERSST | Diff | Std-dev LA-ERSST | Diff | Trend (°C/10yrs) | Diff | Corr |
|---|---|---|---|---|---|---|---|
| 1 | 12.26 | -0.76 | 0.53 | 0.00 | 0.16 | 0.01 | 0.69 |
| 2 | 12.06 | 0.15 | 0.55 | 0.03 | 0.17 | 0.09 | 0.70 |
| 3 | 12.68 | -1.14 | 0.54 | 0.17 | 0.17 | 0.12 | 0.82 |
| 4 | 13.59 | -2.23 | 0.52 | 0.07 | 0.16 | 0.05 | 0.78 |
| 5 | 12.65 | 0.71 | 0.54 | 0.05 | 0.16 | 0.06 | 0.77 |
| 6 | 13.16 | -0.51 | 0.52 | 0.06 | 0.16 | 0.01 | 0.79 |
| 7 | 14.62 | -2.53 | 0.50 | 0.09 | 0.14 | 0.00 | 0.76 |
| 8 | 14.62 | -0.25 | 0.50 | 0.14 | 0.14 | 0.03 | 0.85 |
| 9 | 14.62 | -0.86 | 0.50 | 0.12 | 0.14 | 0.08 | 0.78 |
| 10 | 15.92 | -1.06 | 0.50 | 0.07 | 0.12 | 0.09 | 0.81 |
| 11 | 18.68 | -1.27 | 0.46 | 0.19 | 0.10 | 0.16 | 0.65 |
| 12 | 21.18 | -3.51 | 0.37 | 0.28 | 0.12 | 0.12 | 0.70 |
| 13 | 21.18 | -2.98 | 0.37 | 0.19 | 0.12 | 0.05 | 0.71 |
| 14 | 24.37 | -4.39 | 0.32 | 0.20 | 0.12 | 0.09 | 0.69 |
| 15 | 24.37 | -2.83 | 0.32 | 0.19 | 0.11 | 0.08 | 0.75 |
| 16 | 23.02 | -3.80 | 0.33 | 0.22 | 0.11 | 0.06 | 0.77 |
| 17 | 23.02 | -3.30 | 0.33 | 0.28 | 0.12 | 0.07 | 0.71 |
| 18 | 24.37 | -3.53 | 0.32 | 0.13 | 0.11 | 0.02 | 0.63 |
| 19 | 24.37 | -3.35 | 0.32 | 0.12 | 0.11 | 0.02 | 0.65 |
| 20 | 25.44 | -3.01 | 0.31 | 0.13 | 0.09 | 0.06 | 0.67 |
| 21 | 25.28 | -1.66 | 0.35 | 0.14 | 0.04 | 0.14 | 0.56 |
| 22 | 25.23 | -1.63 | 0.41 | 0.15 | 0.02 | 0.18 | 0.49 |
| 23 | 25.23 | 0.56 | 0.41 | 0.03 | 0.03 | 0.17 | 0.37 |
| 24 | 25.81 | -1.35 | 0.37 | 0.12 | 0.01 | 0.16 | 0.54 |
| 25 | 25.81 | -0.81 | 0.37 | 0.11 | 0.01 | 0.16 | 0.66 |
| 26 | 25.96 | -0.16 | 0.34 | 0.10 | 0.05 | 0.13 | 0.47 |