# Peer review of "Testing the validity of regional detail in global analyses of Sea"

_Ocean Science, 2018_

## Referee Comment (RC1) · Anonymous Referee #1 · 18 Jan 2019

The authors present a comparison of three global SST datasets against a local dataset of long-term temperature observations at a series of coastal stations. They investigate the differences mainly with the help of EOF analysis to understand the structure of the deviations. The study is professionally conducted, the methods are appropriately chosen and the presentation of the results is adequate. However, the paper completely fails to put this piece of work into context. Without the context of what is known, the results appear a bit isolated and it is hard to see how your work contributes to science in your field. I list here what I would like to see added. The introduction lacks a general overview on the present state of knowledge about coastal ocean warming worldwide. It is well-known that the coastal ocean reacts differently to atmospheric temperature

changes than the open sea. Putting this regional study into this context would help to assess its novelty and estimate its importance. Have e.g. previous studies identified the SST difference between coast and open sea at your coast? The methods section lacks a description of the differences between the SST data products. They are just introduced as "black box" data series and then compared to the in-situ measurements one by one. But why are these data products different? If you described how these different datasets were constructed, which data went into them and so on, this might also help to gain some understanding on why the deviations from your LH time series are different. The interpretation of the results does not even try physical explanations. Why is the near-coastal temperature below the open-sea SST products? Is there e.g. coastal upwelling of deep-sea water? Please make clear what the present state of knowledge is. In the end, the question is, what do we see in the differences you identified? Do the LH and the LA data differ because there is a real difference between values and trend at the coast and the open sea, or do they differ because the LA data are simply not good enough, maybe based on too few observations? So, do we identify a physical phenomenon or just artefacts in the data products? Please reflect on whether it is possible from the present state of knowledge to answer this question.

General comments:

English language in this article could be improved. Even if I am not a native speaker, I noticed several places where - "the" should have been inserted or avoided, - singular and plural are mixed up, or - inadequate prepositions were chosen. Copy-editing by a native speaker would probably help.

Temperature differences are given in °C or in °, this should be changed to K.

In those sub-figures where your x axis lists the station acronyms, these are too small to read. You could plot them alternatingly in two rows, like in the attached figure, and/or rotate the labels by 90° to increase the font size.   Specific comments:

L29-33: The grammar in this sentence is not precise. Please correct.

L35: "the different dataset" -> "the choice of dataset"?

L42: larger than what?

L111-112: What is the difference between "no change of the mean" and "zero change"?

L175: which may reflect local effects

L280: The time series of PC2 is not stationary, it rather fluctuates around zero with no prominent long-term trend.

L294-297: You mean it confirms the quality of the LH dataset, not the LA dataset, right? The term "alluding to the quality" is a bit unscientific I from my point of view since its interpretation is not clear. Do you consider the fact that the LH is within the range of the LAs′ variability as a support for the credibility of the LH dataset?
* * *
ZMW | XCS |

QHD  LHT

**Fig. 1.**

---

## Short Comment (SC1) · 6 Feb 2019

This is a very interesting paper improving the ability of SST predicting models. The issue addressed in this study is very important and the authors have done a fine job. The finding is useful and important for climate research and modeling communities, although the present analysis/conclusions might be only fair. However, some points need clarification and I would suggest the following revisions.

1. Please further explain the reasons for determining these 26 special coastal stations and what characteristics they have. For example, why important cities such as Shanghai and other Yangtze River Deltas are excluded. Scholars may be very interested in

the changing laws of these coastal stations. Is it possible to supplement them?

2. When the same method is applied to a variety of different dataset, what is the difference, whether there is a need for major factors, especially for data sources with different lengths of timeseries. I think it should be pointed out.

3. The authors used the HadISST1, ERSST, COBE SST, and NOAA OISST dataset to calculate the long-term trend of SST in the marginal seas, e.g., coastal water of China. It is well known that the SST observations are extremely sparse during the early decades until satellite measurements being available in the 1970s, especially within the marginal seas. Therefore, I think a long-term SST trend using such a dataset is hard to convince. The authors need to explain this in more detail.

Based on the above points, this paper has the potential to be a useful contribution to the literature but will elect revision before publication. I think this paper deserves to be published in the journal after minor revision.

—————————————————————

---

## Referee Comment (RC2) · Anonymous Referee #2 · 15 Feb 2019

General comments:

This paper compares a long, homogenized timeseries of in situ observations of coastal sea surface temperature (SST) with several SST analysis datasets. The basic idea is scientifically sound, the assessment is detailed and the paper is well organised. None of the results are particularly surprising, but it is still useful to have performed the comparison. There are some grammatical errors throughout, so it is recommended that the authors arrange for the language to be reviewed by a native English speaker prior to publication.

The paper requires more information to be provided in the introduction on the SST

analyses used, in order to put the results into proper context in the discussion:

* What are the differences between the analysis products, including input data, expected feature resolution capability etc? What is the depth of each analysis compared to depth of the in situ observations?

* What data do each of the analyses include in this location from the time period prior to the satellite era? Do some of the analyses include data from the same sources? Are the in situ observations used for the assessment definitely independent of the analyses?

* Are there uncertainties included with any of the SST analysis products? What do they look like around the coast?

* Coastal satellite observations of SST are not as reliable as for the open ocean - this should be covered in the discussion.

Additional general comments:

* How is the LH homogenisation applied? Need more detail on how the correction is obtained.

* Using annual means results in removal of a lot of temporal variability. There could be variation of the results in different seasons. Have you looked into this at all?

* If you want to include a comparison to the NOAA OI-SST analysis, this needs a separate results section, rather than presenting it in the conclusions. Similarly, the OI-SST analysis is introduced in the abstract alongside the other analyses, but the same method was not applied to this analysis (similarly in section 2 etc). This needs rewording.

Specific comments:

Table 1: Replace "commonly used" with "used in this study" as there are other datasets available which are also well-used. Include the download date of the datasets too.

Figure 1: Are you able to reproduce this plot if it's already been published? Otherwise need to replot in a different form.

Line 124-125: Need evidence to back up this statement. Also, what are your criteria for "consistent"? Suggest moving some of the information in the Appendix to here.

Line 132: The LH dataset is not an analysis.

Line 134: How are the matchups performed? Is there an interpolation to the observation location?

Line 157: 9 cases is more than "a few", suggest reword.

Figure 2c: Looks like a strong correlation but offset by a bias - elaborate on this.

Line 201: The effect of satellites on the SST analyses is important - this information should be included in the introduction (as mentioned above in general comments as part of the differences we might expect between analyses and in situ dataset).

Lines 202 - 204: Check these values.

Line 208: Elaborate on what is meant by "degenerate" and the implications of this on the results.

Figure 6: What are the anomalies to?

Line 323: Do the SST analyses already attempt to include quality-controlled, homogenized data? (This information should be included in the introduction, see general comments above)

Line 326: There are already several projects dedicated to quality control and homogenization of in situ data - suggest including some information from a literature review here. However, it's also worth including the comment that it is useful to keep some high-quality data separate from that available for analyses, for validation activities such as this one.

---

## Referee Comment (RC3) · Igor Belkin (Referee) · 28 Feb 2019

Review of Yan et al. "Testing the validity of regional detail in global analyses of sea surface temperature — the case of Chinese coastal waters" submitted to the Ocean Science (OS-2018-137)

This is an important paper and it should be accepted and published pending a moderate revision. The manuscript (MS) is written clearly. The data analysis and discussion are succinct and concise. There are a few major concerns and several minor comments/edits.

[Figure]

Major comments:

This MS presents results that appear striking. Systematic differences between locally measured annual mean SST (termed "locally homogenized" or LH SST) and gridded analyses (LA) amount to 4°C (!) as illustrated by the remarkable Fig. 7 that shows SST differences (at 26 coastal locations) between LH and LA-OISST, LH and LA-ERSST, LH and LA-COBE SST, and LH and LA-HadISST. All four gridded analyses (LA) are warmer than LH at all (26) but one coastal station. The differences between local SST and gridded SST (denoted here LH and LA for brevity) peak off the SE China, along the Taiwan Strait coast, between Putian in the north, Xiamen in the middle, and Dongshan Island in the south.

Alas, the authors, while documenting their results quite extensively, have not invested enough efforts in analyzing and explaining such striking results. Moreover, they seem to downplay the obvious importance of their own results. Otherwise, it is hard to explain why these major discrepancies are not mentioned in the abstract and only briefly mentioned in the end of Section 5 ("Discussion and conclusions").

The authors are probably correct when they state (line 302) that these differences are "related to the coastal position of LH, and the averaging in the LA." Yet they do not offer any physical mechanism that would explain such large differences between coastal and offshore SST. The most important and plausible mechanisms are (1) winter cooling (exceeding 2°C along the Zhejiang-Fujian coast) on the inshore side of the Zhejiang-Fujian front (Hickox et al., 2000) and (2) southward transport of cold water by the China Coastal Current (CCC). As pointed out by Belkin and Lee (2014, p. 830), "The crucial role of CCC in regulating the hydrography of the Taiwan Strait was amply demonstrated by the 2008 cold disaster caused by southward invasion of the CCC waters, when SST dropped by almost 8 °C vs. a 12-year mean February SST." Hopefully, the authors will address these mechanisms in a follow-up paper that should focus on seasonal variability of the above-mentioned discrepancies between LH and LA.

Minor comments:

(1) I have highlighted quite a few passages in the text that should be re-worded. The authors are able to improve the text themselves. Therefore, I have not suggested any specific edits. The annotated manuscript is uploaded. (2) In Fig.1, stations should be shown with consecutive numbers (as in Table 2), not acronyms. (3) Coordinates (lat, lon) of all 26 stations should be documented in the paper. (4) Perhaps, the text would be easier to read should the authors cite full names of 26 stations (complete with their numbers) vs. 26 acronyms. I would argue that full names are easier to memorize than respective acronyms, especially when the full names are accompanied by respective numbers. For example, it is easy to remember that there is a large spatial gap between Station 10 (LYG) and 11 (SPU) (Fig. 1 and Table 2).

Conclusion: I recommend acceptance pending moderate revision focused on adding a thorough discussion of physical mechanisms responsible for discrepancies between LH and LA.

* * *
[Figure]

**Supplement:**

[revised manuscript text omitted]

---

## Author Comment (AC1) · 29 Mar 2019

Dear Referee:

Thank you very much for the helpful comments on our manuscript "Testing the validity of regional detail in global analyses of Sea surface temperature-the case of Chinese coastal waters" (No: os-2018-137). For the revision, we fully considered all suggestions and give the item-by-item reply. Also, we tried our best to improve the English writing in our manuscript. And revised portion are highlighted in yellow in the manuscript. We appreciate for the Reviewer's work and hope that the correction will meet with approval. Once again, thanks very much for your comments and suggestions.

[Figure]

Best regards

Yan Li, Hans von Storch, and coauthors

Major comments

1. The introduction lacks a general overview on the present state of knowledge about coastal ocean warming worldwide.

Reply: Following with the comment, we have added a general overview on the coastal ocean warming worldwide. The new added context is as follows: A study of SST changes in the world ocean with large marine ecosystems revealed that the Subarctic Gyre, European Seas, and East Asian Seas warmed at rates 2-4 times the global mean rate (Belkin 2009). Recently, Lima and Wethey 2012 using a SST dataset with higher spatial-temporal resolution detected that during the last three decades $\sim 71.6\%$ of the world coastal locations have experienced a warming trend of $0.25\pm0.13°$C per decade and 6.8% a cooling of $-0.11\pm0.10°$C per decade. Increase in SST is especially important in coastal areas due to its severe impact in coastal ecosystems (Honkoop et al., 1998; Burrow et al., 2011; Wernberg et al. 2016).

2. Have e.g. previous studies identified the SST difference between coast and open sea at your coast?

Reply: Following with the comment, we have added conclusions of previous studies in the revision: As part of the Northwest Pacific Ocean, the marginal China Seas are located at one of the largest continental shelves in the world, with many coastal up-welling currents (Yan, 1992; Guan 2009; Wang et al., 2012, Xie et al., 2016). Upwelling can cause the upward movement of sea water from deeper layer into the surface layer and cool the SST at the upwelling region. Pohlmann (1987) found a negative surface temperature anomaly along the western South China Sea (SCS) from Gulf of Tokin to the central Vietnam in summer. Forced by a strong westerly monsoon during the 2000 cruise, the maximum upwelling with the coldest water cooler than 23°C was centered

off Shantou city of China (Guan 2009). Owing to the coastal upwelling, SST in the coast of the eastern Hainan was 1∼2°C lower than ambient offshore water and SST in the Yangzte River Estuary was 2-3°C lower than ambient offshore water; 10m sea temperature in the coast between eastern Guangdong and southern Fujian provinces was lower than surrounding sea water about 5°C (Zhao et al., 2001; Xu et al., 2014; Xie et al., 2016).

3. Why is the near-coastal temperature below the open-sea SST products?

Reply: Our study informs that the near-coastal SSTs are below the open-sea SST products which may due to coastal upwelling in China Seas. And we added the following sentences to further explain the point in the revision: In the China Seas, most of the coastal upwelling currents occur at the ECS and the northern SCS, other small upwelling currents at the tops of the Liaodong Peninsula and Shandong Peninsula (Figure 1). The consensus of previous studies is that coastal upwelling currents results in cooling SST at these coastal areas (Xie S P, 2003; Guan et al., 2009; Su et al., 2012). In our study, we find that the in situ shoreline SSTs at the upwelling areas (e.g. Laohutan station, Shidao station and Dongshan station) are always colder than global gridded SST data, with the value of below -1°C (Table 2, Table 3 in the revision). We hypothesize that these negative differences are connect with coastal upwelling. To test this hypothesis, we examine the output of a numerical simulation of the currents in the South China Sea with a grid resolution of. 0.04°. The model is embedded in an almost global model with 1° grid resolution (Tang et al., 2018). The model used is Hybrid Co-ordinate Ocean Model (HYCOM) that is exposed to periodic climatological atmospheric forcing, with a fixed annual cycle but no weather disturbances. The atmospheric forcing comes from the Comprehensive Ocean-Atmosphere Data set (COADS). We extract simulated SSTs at three different distances (near the station, 50km, and 100km from each coastal hydrological station in SCS). Figure 7 in the revision shows that most shoreline SSTs are lower than ambient offshore SSTs, especially SSTs at 100km from shoreline. However, the stations Beihai (No.22) and Weizhou (No.23) are not affected

by coastal upwelling, and consistently, there are no notable differences among SSTs at three different distances from the two stations. The result reflects that the homogenized SST data set for shoreline stations catch this relative cooling water effect of the regional upwelling currents. On the other hand, the global gridded SST datasets point to higher temperatures which may be caused by their coarse resolution or by the lack of near-shore observations when compiling near-shore box averages in coastal areas (Wang et al., 2018).

Figure 1. Study area and locations of 26 coastal sites (a), for which continuous monthly SST recordings are available and corrected by eliminating inhomogeneities. The identified breakpoints in individual SST stations from 1960-2015 (b).Results from Li et al. 2018. Black circle represents 26 coastal sites and blue arrow represents coastal upwelling.

Figure 2. Simulated SSTs at different distances from each coastal hydrological station in SCS.

General comments:

(1) English language in this article could be improved. Even if I am not a native speaker, I noticed several places where - "the" should have been inserted or avoided, - singular and plural are mixed up, or - inadequate prepositions were chosen. Copy-editing by a native speaker would probably help.

Reply: We made our best to improve the language in the revised manuscript. The changes made will not influence the content and framework of the paper. Here we have not listed the language changes but all of them have been marked in yellow color in the revised manuscript.

(2) Temperature differences are given in °C or in °, this should be changed to K.

Reply: the temperature differences are changed into K in the revised paper.

(3) In those sub-figures where your x axis lists the station acronyms, these are too

small to read. You could plot them alternatingly in two rows, like in the attached figure, and/or rotate the labels by 90 to increase the font size.

Reply: We have redrawn all of the figures in our paper to improve the quality. Combing with the second referee's comment, we modify figures which x axis lists the station No., so as to increase the font size. The modified figures (e.g. Figure 3 and Figure 7) can be found in the revised manuscript.

Figure 3. Comparison of the EOF1 and EOF2 derived from the LH data set of local SST at 26 sites (blue bars; red lines), and derived from the localized analysis data LA-HadISST (yellow bars; black lines). Top: EOF spatial patterns, bottom: principal components (time coefficients).

Figure 7. The mean SST differences at the 26 locations between LH and LA-OISST (1982-2015; red line), LH and LA-ERSST (1960-2015; blue line), LH and LA-COBE SST (1960-2015; green line) and LH and LA-HadISST (1960-2015; black line)

Specific comments:

L29-33: The grammar in this sentence is not precise. Please correct.

Reply: L29-33 "A number of extended historical observed SST products have been used in global climatological community (Boehme et al. 2014; Hirahara et al. 2014), as well as in the regional climate change, for example the China Seas, the Baltic Sea and North Sea (Belkin, 2009; Wu et al., 2012; Stramska and Bialogrodzka, 2015)." is modified as "Long-term historical SST data sets have been extensively used as a source of information on global and regional SST trends and variability (Belkin, 2009; Wuet al., 2012; Boehme et al.2014; Hirahara et al.2014; Stramska and Bialogrodzka, 2015)." in the revised paper.

L35: "the different dataset" -> "the choice of dataset"?

Reply: We replaced the term "the different dataset" to "the choice of dataset" in the revised paper.

L42: larger than what?

Reply: We replaced "larger" by "large" in the revised.

L111-112: What is the difference between "no change of the mean" and "zero change"?

Reply: we replaced L111-112 "At most of the 26 stations, a downward correction has been found necessary – only at two stations (Yunwo and Pingtan) an upward change was stipulated, in one case no change of the mean and in the remaining 23 a downward or zero change." by "At 22 of the 26 stations, a downward correction of the mean has been found necessary – only at two stations (Weizhou and Pingtan) an upward change was stipulated, and in two case nearly no change of the mean (Naozhou and Shidao)" in the revised paper.

L175: which may reflect local effects

Reply: L175 is modified "The local data indicate markedly lower temperatures, which may reflect by local effects" to "The local data indicate markedly lower temperatures, which may mainly be because of coastal upwelling, but also other local effects, including local tidal mixing, sea front, sea water vertical mixing, and fresh water discharge, etc." in the revised paper.

L280: The time series of PC2 is not stationary, it rather fluctuates around zero with no prominent long-term trend.

Reply: No, "stationarity" does not imply constancy but that there is no change of the statistical properties in time. But for avoiding misunderstanding, we replaced "PC2 is mostly stationary" by "The time series of PC2 fluctuates around zero without prominent long-term trend".

L294-297: You mean it confirms the quality of the LH dataset, not the LA dataset, right? The term "alluding to the quality" is a bit unscientific I from my point of view since its interpretation is not clear. Do you consider the fact that the LH is within the range of the LAs' variability as a support for the credibility of the LH dataset?

Reply: Yes, it is support for the LH data set; "Alluding" is a proper English term, but for avoiding irritations, we replaced it by "points to".

References:

Belkin, I.M. Rapid warming of large marine ecosystem, Prog. Oceanogr., 81, 207-213, 2009.

Burrow, M. T., et al. The pace of shifting climate in marine and terrestrial ecosystems, Science, 334, 652-655, 2011.

Honkoop R.J.C., der Meer, J.Van, Beukema, J. J. and Kwast D. Does temperature-influenced egg production predict the recruitment in the bivalve Macoma Balthica? Mar. Ecol. Prog. Ser., 64, 229-235,1998.

Guan, J., Cheung, A., Guo, X. and Li, L. Intensified upwelling over a widened shelf in the northeastern South China Sea, J. Geophys. Res., 114, C09019, doi:10.1029/2007JC004660, 2009.

Su, J., Xu, M., Pohlmann, T., Xu D., Wang D. A western boundary upwelling system response to recent climate variation (1960–2006), Cont. Shelf Res., 57(2013)3-9, 2012.

Lima, F.P. and Wethey, D.S. Three decades of high-resolution coastal sea surface temperatures reveal more than warming, Nat. Commun., 3,704, 2012.

Tang, S., von Storch, H. Chen, X., and Zhang, M. "Noise" in climatologically driven ocean models with different grid resolution, Oceanologia, 10.1016/j.oceano.2019.01.001, 2019.

Wang, Q.Y., Li, Y., Li, Q.Q., et al. A comparison and evaluation of two centennial-scale sea surface temperature datasets in the China Seas and their adjacent sea areas, J. Trop. Meteor., 24(4), 452-460, 2018.

Wernberg, T., Bennett, S., Babcock, R.C., et al. Climate-driven regime shift of a temperature marine ecosystem, Science, 353, 169−172, 2016.

[Figure]

Xie, S.P., Xie, Q., Wang, D., and Liu, W.T. Summer upwelling in the South China Sea and its role in regional climate variations, J. Geophys. Res., 108(C8), 3261, doi:10.1029/2003JC001867, 2003.

Zhao, B.R., Ren, G.F., Cao, D.M., Yang, Y.L. Characteristics of the ecological environment in upwelling area adjacent to the Changjing River Estuary, Oceanol. Limnol. Sin., 32(3), 327-333, 2001. (in Chinese with English abstract)

Yan, T.Z. A preliminary classification of coastal upwellings in the China Seas. Mar. Sci. Bull., 10(6), 1-6, 1991. (in Chinese with English abstract)

Please also note the supplement to this comment:
https://www.ocean-sci-discuss.net/os-2018-137/os-2018-137-AC1-supplement.pdf

---

## Author Comment (AC2) · 29 Mar 2019

Dear Prof.Wu:

Thank you very much for the helpful comments on our manuscript "Testing the validity of regional detail in global analyses of Sea surface temperature - the case of Chinese coastal waters" (No: os-2018-137). For the revision, we fully considered all suggestions and give the item-by-item reply. Also, we tried our best to improve the English writing in our manuscript. And revised portion are highlighted in yellow in the manuscript. We appreciate for the your work and hope that the correction will meet with approval. Once again, thanks very much for your comments and suggestions.

Best regards

Yan Li, Hans von Storch, and coauthors

1. Please further explain the reasons for determining these 26 special coastal stations and what characteristics they have. For example, why important cities such as shanghai and other Yangtze River Deltas are excluded. Scholars may be very interested in the changing laws of these coastal stations. Is it possible to supplement them?

Reply: Thank you for pointing it out. And we have added this explanation in the revised paper. Currently, more than 100 hydrological stations are operating and monitoring near-shore hydrological conditions. Among these stations, only 26 stations have routinely and continuously recorded since 1960, with a percentage of missing data lower than 4%, Also, these stations have undergone only a few (five and less) and documented relocations. Among the 26 stations, there are only few along the southern Yellow Sea Sea, because this area is a vast muddy coast which is not suitable for hydrological stations. Unfortunately, the limited data sets for this region do not satisfy our needs of temporal coverage and completeness of records: Since 2000s, there have been some automatic stations. Around the city of Shanghai, there are 7 hydrological stations (Figure 1). Among them, only the Tanhu station has at least 50 yr of continuous observations of SST in the period from 1950 to 2015 (66 years) (Table 1). However, percentage of missing data of SST series in Tanhu station is higher than 4%. Therefore, our data set has no entries for the Shanghai/Yangtze River Deltas area.

Figure 1. 7 hydrological stations around Shanghai

2. When the same method is applied to a variety of different dataset, what is the difference, whether there is a need for major factors, especially for data sources with different lengths of time series. I think it should be pointed out.

Reply: In this study, the same method was applied to a variety of different datasets. We found that all of these globally gridded datasets exhibit broadly the same pattern in

space and time as the EOFs of the local homogenized (LH) SST data from 26 coastal sites. However, all of these globally gridded datasets exhibit surface temperatures usually higher than the LH data. This difference may be caused by two factors. First, there are several the coastal upwelling currents at the East China Sea and the northern South China Sea, leading to cooler water temperature than nearby. The cool SST hardly can be described in the globally gridded datasets due to their coarse spatial resolution. Second, we suggest that this is related to the coastal position of LH, and the averaging in the LA data. The differences are largest in the case of the coarsest analysis (ERSST), but weakest in the OISST-data set with a resolution of a quarter of a resolution degree.

3. The authors used the HadISST1, ERSST, COBE SST, and NOAA OISST dataset to calculate the long-term trend of SST in the marginal seas, e.g., coastal water of China. It is well known that the SST observations are extremely sparse during the early decades until satellite measurements being available in the 1970s, especially within the marginal seas. Therefore, I think a long-term SST trend using such a dataset is hard to convince. The authors need to explain this in more detail.

Reply: We agree with you, and this is one of our main results, namely that the homogenized local data (LH) are partly inconsistent with the variability and trends in the global analyses (LA). We are not showing the trends in the LA-SST because of a wish to documents the real trends, but to assert if the LS-SST trends (and variability) are consistent with the LH trends and variability. The result is that there are inconsistencies. This is explained in the manuscript.

Please also note the supplement to this comment:
https://www.ocean-sci-discuss.net/os-2018-137/os-2018-137-AC2-supplement.pdf

**Supplement:**

[Figure]

Figure 1. 7 hydrological stations around Shanghai

---

## Author Comment (AC3) · 29 Mar 2019

Dear Referee: Thank you very much for the helpful comments on our manuscript "Testing the validity of regional detail in global analyses of Sea surface temperature - the case of Chinese coastal waters" (No: os-2018-137). For the revision, we fully considered all suggestions and give the item-by-item reply. Also we try our best to improve the English writing in our manuscript. And revised portion are highlighted in yellow in the manuscript. Once again, thanks very much for your comments and suggestions.

Best regards

[Figure]

Yan Li, Han von Storch, and coauthors

Major Comments:

1. What are the differences between the analyses products, including input data, expected feature resolution capability etc? What is the depth of each analysis compared to depth of the in situ observations?

Reply:

(1) There are some differences of data sources, bias adjustment and reconstruction method, etc. in the SST analyses products. Some analyses only use in situ observations, such as ERSSTv4 and COBE SST. Others use both in situ and satellite observations, such as OISST and HadISST. Details are shown as follows: 1) HadISST dataset. HadISST dataset is composed of the SST data from Marine Data Bank in the United Kingdom (mainly ship tracks) and the International Comprehensive Ocean-Atmosphere Data Set (ICOADS). ICOADS is mainly from ships, buoys, automated platform types, moored buoys, drifting buoys, and near-surface measurements from hydrographic profiling studies. From 1982 onward, HadISST also consists of adjusted satellite-derived SST data from the AVHRR. Though data gaps in HadISST1 have been interpolated, the performance of HadISST1 in describing the SST variability in the China Seas, particularly in the inshore areas, is still questionable, because of the data sparseness and limitations of the interpolation techniques (Rayner et al. 2003; Zhang et al. 2005; Liu et al. 2015; Li et al., 2017). 2) COBE SST dataset. COBE SST also composes the ICOADS, U.K. Marine Data Bank, and U.S. Marine Meteorological Journals (Hirahara et al. 2012). To construct unobserved variability in data-sparse regions, satellite observations are incorporated into the present objective analysis scheme. However, the satellite observations are used only for constructing empirical orthogonal functions (EOF) that represent interannual-to-decadal SST variations; they are not used in the final COBE SST to preserve the homogeneity of the SST analysis for more than 100yrs. Hirahara et al. (2014) pointed out that this was because the use of satellite data for

the whole analysis makes the grid wise variability of SST analysis larger by 10%-20%, compared with that without the data. 3) ERSST v4 dataset. The historical ocean observations used for ERSST v4 analysis arise from the in situ ICOADS from 1854 to 2007, and from the Global Telecommunication System (GTS) receipts from NCEP after 2007. The ICOADS and GTS observations exhibit both random errors and systematic biases (Kennedy et al., 2011). Huang et al. (2016) pointed out that filters and EOF decompositions were used to reduce the effect of the random errors, and bias adjustments are applied to remove the systematic biases in the ERSST v4 analysis. 4) OISST dataset. This data set is the Optimum Interpolation (OI) SST Analysis Product, which uses Advanced Very High Resolution Radiometer infrared satellite SST data from the Pathfinder satellite combined with buoy data, ship data, and sea ice data SST data sets. In order to apply the correction for bias in OISST, the satellite data have been classified into daytime nighttime bins and corrected separately using the patterns of 15 day averaged in situ SSTs by NOAA's OI algorithm. The bias-corrected daytime and nighttime satellite SST, ship, and buoy SSTs are merged based on noise-to-signal ratio maps for each data type, which have averaged weights of 15.1, 15.1, 1.0, and 15.1, respectively. Therefore it can be interpreted as the bulk SST at about 0.5 m depth (Reynolds et al., 2007).

(2) The depths

The measurement depths of SST sensors vary because there is an abundance of data sources in the ICOADS. For example, Bulk carriers, vehicle carriers, gas tankers, and livestock carriers typically measure SST at a 7-m depth or deeper (Kent and & Taylor, 2006). Research vessels, fishing vessels, trawlers, support vessels, the Coast Guard, and sailing vessels all typically measure SST at a 4-m depth or shallower (Kent et al. 2007). However, these SST observations above are much sparser along the shoreline of China (Li et al., 2017). SST observations from hydrological stations of China are the water temperature at the depth of 0.5~1 meter below sea level (Li et al., 2018). Thus, it is very difficult to address all of the issues related to variations in observing

observation systems, changes in measurements, and depths that can lead to potential errors (Kent and & Woodruff, 2006, Woodruff et al. 2011).

2. What data do each of the analyses include in this location from the time period prior to the satellite era? Do some of the analyses include data from the same sources? Are the in situ observations used for the assessment definitely independent of the analyses?

Reply: The 26 coastal hydrological stations at the coastline of China have been taking routine observations since 1960, with few missing data. All of these in-situ SST data from 1960 to 2015 are provided by the National Marine Data and Information Service (NMDIS) of China and have been quality controlled and homogenized recently by Li, et al., (2018). These SSTs data from coastal hydrological stations have never been merged into HadISST, COBE SST or other gridded SST analyses. Therefore, the homogenized long-term SST observations along the Chinese coast can be used for evaluation on these analyses.

3. Are there uncertainties included with any of the SST analysis products? What do they look like around the coast?

Reply: As we mentioned above, the main data source of these SST analyses is ICOADS SST. However, the spatial distribution of the data density and coverage of this data set vary in the ICOADS because of the uneven distribution of ship routes and the existence of several data-sparse regions (Woodruff et al. 2011). To understand the data density and coverage of the observations over the China Seas and their adjacent waters, the numbers of SST observations from the ICOADS R2.5 (which are in the International Maritime Meteorological Archive (IMMA) format covering) are counted up for each 1° grid over this region in our previous work (see Figure 1 in Li et al., 2017). The numbers of SST observations from ICOADS R2.5 are not well distributed over the China Seas, especially in the Bohai Sea and the Yellow Sea. The data density is much sparse in these areas. Thus, the limited data coverage of the inshore areas can lead

to high uncertainties in the estimates of SST variability in these regions.

Reference: Yan Li, Lin Mu, Yulong Liu, et al. 2017. Analysis of variability and long-term trends of sea surface temperature over the China Seas derived from a newly merged regional data set. Climate Research, 73: 217-231.

4. Coastal satellite observations of SST are not as reliable as for the open ocean – this should be covered in the discussion.

Reply: Yes, we agree with it and added a comment in the revision; however this is hardly of significance for our study.

5. How is the LH homogenisation applied? Need more detail on how the correction is obtained.

Reply: Sea surface temperature (SST) measurements from 26 coastal hydrological stations of China had been homogenized and analyzed in our previous work (Li et al., 2018). For avoiding repetition, we simply summarize the homogeneity process in the revised paper, that is, "Monthly mean SST series were then derived and subjected to a statistical homogeneity test, called the Penalized Maximum T (PMT) test (more details can be found in Li et al., 2018). Homogenized monthly mean SST series were obtained by adjusting all significant change points which were supported by historic metadata information".

6. Using annual means results in removal of a lot of temporal variability. There could be variation of the results in different seasons. Have you looked into this at all?

Reply: In our work, we consider annual mean values. Some analyses with seasonal mean values are also calculated, but these are not covered by our present account and merely summarized.

7. If you want to include a comparison to the NOAA OI-SST analysis, this needs a separate results section, rather than presenting it in the conclusions. Similarly, the OI-SST analysis is introduced in the abstract alongside the other analyses, but the

same method was not applied to this analysis (similarly in section 2 etc). This needs rewording.

Reply: Thanks for your suggestion. The fourth SST product, OISST uses Advanced Very High Resolution Radiometer infrared satellite SST data from the Pathfinder satellite combined with buoy data, ship data, and sea ice data, covering from 1982 to present. Due to its high spatial resolution of 0.25°x0.25°, it is used in the concluding section for clarifying some additional aspects, such as the global gridded SST datasets point to higher temperatures which may be caused by their coarse resolution. Following the comment, we modified the abstract, introduction and discussion in the revised paper.

Specific comments:

1. Table 1: Replace "commonly used" with "used in this study" as there are other datasets available which are also well-used. Include the download date of the datasets too.

Reply:" Table 1. Global gridded SST datasets that are commonly used for climate studies" is modified as "Table 1. Global gridded SST datasets that are used in this study".

2. Figure 1: Are you able to reproduce this plot if it's already been published? Otherwise need to replot in a different form.

Reply: Fig.1 has been replotted in a different form. Actually, the identified breakpoints information in Li et al (2018) was shown only in Table 2.

3. Line 124-125: Need evidence to back up this statement. Also, what are your criteria for "consistent"? Suggest moving some of the information in the Appendix to here.

Reply: The information about "The consistency of homogenized SST data set with homogenized SAT data set" in the Appendix A is moved into Section 3.

[Figure]

4. Line 132: The LH dataset is not an analysis. Line 134: How are the matchups performed? Is there an interpolation to the observation location?

Reply: "Local homogenized SST-analysis" is modified as "Local homogenized SST" in the revised paper.

5. Line 157: 9 cases is more than "a few", suggest reword.

Reply: "the standard deviations are in most cases (17) larger for LH, and only in few cases (9) smaller." is modified as "65.4% of the standard deviations (17) are larger for LH, and 34.6% cases (9) smaller."

6. Figure 2c: Looks like a strong correlation but offset by a bias - elaborate on this.

Reply: The local data indicate markedly lower temperatures, which may mainly be because of coastal upwelling. In the China Seas, most of the coastal upwelling currents occur at the ECS and the northern SCS, other small upwelling currents at the tops of the Liaodong Peninsula and Shandong Peninsula (Figure 1). The consensus of previous studies is that coastal upwelling currents results in cooling SST at these coastal areas (Xie S P, 2003; Guan et al., 2009; Su et al., 2012). Owing to the coastal upwelling, SST along the coast of the eastern Hainan was 1∼2°C lower than ambient offshore water, and SST in the Yangzte River Estuary was 2-3°C lower than ambient offshore water; 10m sea temperature along the coast between eastern Guangdong and southern Fujian provinces was lower than surrounding sea water about 5°C (Zhao et al., 2001; Xu et al., 2014; Xie et al., 2016). Station 15 (Pingtan) to Station 19 (Yunwo) along the East China Sea coast are at the upwelling areas. In our study, we find that the in situ shoreline SSTs from Station 15 to Station 19 are colder than global gridded SST data, with the value of below -2°C.

7. Line 201: The effect of satellites on the SST analyses is important - this information should be included in the introduction (as mentioned above in general comments as part of the differences we might expect between analyses and in situ dataset).

[Figure]

Reply: Reliable SST retrievals from satellites start in in the early 1980s while in situ observations are available much earlier but have changed substantively through time in both their methods of measurement and where the measurements are taken. Since 1980s the Advanced Very High Resolution Radiometer infrared satellite SST data from the Pathfinder satellite are available. These data improve SST sampling, especially in the Southern Ocean and coastal areas (Smith et al., 2008; Lima and Wethey 2012). These data are incorporated into HadISST and OISST combined with buoy data, ship data, and sea ice data SST data sets.

8. Lines 202 - 204: Check these values.

Reply: the values are correct. But for avoid misunderstanding, "Fig. 3d; this corresponds to a mean difference of 0.04K at the southern stations during that time, and a mean difference 0.04K at the northern stations (Fig. 3b)" is modified as "Fig. 3d; this corresponds to a mean difference of 0.04K at the southern stations from Stations 11-26 during that time, and a mean difference 0.04K at the northern stations from Stations 1-10 (Fig. 3b)"

9. Line 208: Elaborate on what is meant by "degenerate" and the implications of this on the results.

Reply: "degenerate" is a technical term, which is well defined and explained in the relevant literature. It is related to the problem with multiple eigenvalues. For details refer to the textbook of von Storch and Zwiers (1999) or other literature on the "significance" of EOFs.

10. Figure 6: What are the anomalies to?

Reply: "Figure 6. Spatial variability of the EOF1 (a); EOF2 (b) mode of the differences between LH anomalies and LA-ERSST anomalies (LH anomalies minus LA-ERSST anomalies). And the time coefficient series of PC1(c) and PC2 (d) from 1960 to 2015." is modified as "Figure 6. EOF analysis of the differences LH-LA-ERSST: Top: EOF

spatial patterns (EOFs), bottom: principal components (time coefficients)."

11. Line 323: Do the SST analyses already attempt to include quality-controlled, homogenized data? (This information should be included in the introduction, see general comments above)

Reply: The quality-controlled homogenized SST data have not been included in the SST gridded analyses. We add this information into the revised introduction.

12. Line 326: There are already several projects dedicated to quality control and homogenization of in situ data - suggest including some information from a literature review here. However, it's also worth including the comment that it is useful to keep some high-quality data separate from that available for analyses, for validation activities such as this one.

Reply: Following with the comment, we have added conclusions of previous studies in the revision: There are several projects or researches dedicated quality control and homogenization of in situ data (Kuglitsch et al., 2012; Hausfather et al., 2016; Minola et al., 2016). It is useful to keep some high-quality data separate from that available for analyses, for validation activities such as our work and others' work (Hausfather et al., 2017).

References:

Hausfather Z and Coauthors, 2017. Assessing recent warming using instrumentally homogeneous sea surface temperature records. Sci. Adv., 3, 31601207, doi:10.1126/sciadv.1601207.

Hausfather, Z., K. Cowtan, M. J. Menne, and C. N. Williams Jr. (2016), Evaluating the impact of U.S. Historical Climatology Network homogenization using the U.S. Climate Reference Network, Geophys. Res. Lett., 43, 1695–1701, doi:10.1002/2015GL067640.

Minola, L., Azorin-Molina, C., & Chen, D. L. (2016). Homogenization and assessment

of observed near-surface wind speed trends across Sweden, 1956-2013. Journal of Climate, 29(20), 7397-7415. https://doi.org/10.1175/JCL1-D-15-0636.1.

Kuglitsch, F.G., Auchmann, R., Bleisch, R., Bronnimann, S., Martius, O., & Stewart, M. (2012). Break detection of annual Swiss temperature series. Journal of Geophysical Research, 117(D13105), 1-12. https://doi.org/10.1029/2012JD017729.

---

## Author Comment (AC4) · 29 Mar 2019

Dear Prof.Belkin:

Thank you very much for the helpful comments on our manuscript "Testing the validity of regional detail in global analyses of Sea surface temperature - the case of Chinese coastal waters" (No: os-2018-137). For the revision, we fully considered all suggestions and give the item-by-item reply. Also we try our best to improve the English writing in our manuscript. And revised portion are highlighted in yellow in the manuscript. We appreciate for your work. Much thanks for your comments and suggestions.

[Figure]

Best regards

Yan Li, Hans von Storch, and coauthors

Major Comments:

1. Why there are major discrepancies between LH and LA?

Reply: Thank you for pointing this out. Our study informs that the near-shore SSTs are below the open-sea SST products which may partly due to the effect of coastal upwelling. And we added the following sentences to further explain the point in the revision:

In the China Seas, most of the coastal upwelling currents occur at the ECS and the northern SCS, other small upwelling currents at the tops of the Liaodong Peninsula and Shandong Peninsula (see Figure 1 in the revision) (Yan 1991). The consensus of previous studies is that coastal upwelling currents results in cooling SST at these coastal areas (Xie et al, 2003; Guan et al., 2009; Su et al., 2012). In our study, we find that the in situ shoreline SSTs at the upwelling areas (e.g. Laohutan station, Shidao station and Dongshan station) are always colder than global gridded SST data, with the value of below -1°C (see Table 2, Table 3 in the revision).

We hypothesize that these negative differences are connect with coastal upwelling. To test this hypothesis, we examine the output of a numerical simulation of the currents in the South China Sea with a grid resolution of. 0.04°. The model is embedded in an almost global model with 1° grid resolution (Tang et al., 2018). The model used is Hybrid Coordinate Ocean Model (HYCOM) that is exposed to periodic climatological atmospheric forcing, with a fixed annual cycle but no weather disturbances. The atmospheric forcing comes from the Comprehensive Ocean-Atmosphere Data set (COADS). We extract simulated SSTs at three different distances (near the station, 50km, and 100km from each coastal hydrological station in SCS). Figure 7 in the revision shows that most shoreline SSTs are lower than ambient offshore SSTs, especially SSTs at 100km from shoreline. However, the Stations 22 (Beihai) and Station 23 (Weizhou) are not affected by coastal upwelling, and consistently, there are no notable differences among SSTs at three different distances from the two stations.

The result reflects that the homogenized SST data set for shoreline stations catch this relative cooling water effect of the regional upwelling currents. On the other hand, the global gridded SST datasets point to higher temperatures which may be caused by their coarse resolution or by the lack of near-shore observations when compiling near-shore box averages in coastal areas (Wang et al., 2018). Besides, there still some other local mechanisms with smaller scale can cause cooling water in the China Seas, such as China Coastal Current (CCC) (Belkin and Lee, 2014) and Ocean Fronts (Zhao, 1987; Ryan et al., 2000). In them, the shallow water shelf front and estuarine plume front are two major fronts in the Bohai Sea and the Yellow Sea at summer. Coastal current front, upwelling front as well as strong west boundary current front usually appear in the East China Sea and the South China Sea which may also have relationship with coastal upwellings (Feng 2000).

Minor comments:

(1) I have highlighted quite a few passages in the text that should be re-worded. The authors are able to improve the text themselves. Therefore, I have not suggested any specific edits. The annotated manuscript is uploaded.

Reply: We did our work to improve the language in the revised manuscript and these changes not influence the content and framework of the paper. Here we have not listed the language changes but all of them have been marked in yellow color in the revised manuscript.

(2) In Fig.1, stations should be shown with consecutive numbers (as in Table 2), not acronyms.

Reply: Thank you for pointing it out. Following with the comment, we display the 26

stations with consecutive numbers and the full names (see Figure 1 in the revised paper). And black circle represents the locations.

(3) Coordinates (lat, lon) of all 26 stations should be documented in the paper.

Reply: Sorry, the accurate latitude and longitudes of hydrology observational stations in China may not be made public. Instead, we show the distribution of these stations in the Figure 1.

(4) Perhaps, the text would be easier to read should the authors cite full names of 26 stations (complete with their numbers) vs. 26 acronyms. I would argue that full names are easier to memorize than respective acronyms, especially when the full names are accompanied by respective numbers. For example, it is easy to remember that there is a large spatial gap between Station 10 (LYG) and 11 (SPU) (Fig. 1 and Table 2).

Reply: Thank you for your suggestion. We have cited full names of 26 stations in the revised paper.

---

## Author Response (AR2)

**Point-by-point responses to the reviews**

**Reply to Referee #3 Prof. Igor Belkin    Report #1**

1. The authors claim that the coordinates of coastal stations where SST measurements have been made cannot be made public. I have an issue with this claim. The Chinese Government has made huge strides toward data openness and data access. I receive lots of data from China that have been inaccessible decades ago. It's absolutely inconceivable that coastal stations' coordinates will be kept secret these days. The authors must publish the stations' coordinates. This is critically important to their study as some station data can be heavily compromised by the station's proximity to warm effluents, for example. Without knowing precise location of each station it would be impossible to evaluate possible contamination of station data by nearby effluents. The veracity and credibility of this entire study is at stake.

**Reply**: Much thanks for pointing this out. As you mentioned, the Chinese Government has made huge strides toward data openness and data access. We have built the National Marine Science Data Center, National Science & Technology Resource Sharing Service Platform of China ([http://mds.nmdis.org.cn](http://mds.nmdis.org.cn)). In the platform, many kinds of marine science data have been shared, including the marine observational data, reanalysis data, statistical data, etc. Currently, more than 100 coastal hydrological stations are operating and monitoring near-shore hydrological conditions. However, not all of these stations information and their observed data are publicly available according the rule of State Oceanic Administration of China. In our study, there are only 9 stations (i.e., Xiaochangshan (39.2 °N, 122.7 °E), Laohutan (38.9 °N, 121.7 °E), Yantai (37.6 °N, 121.4 °E), Xiaomaidao (36.0 °N, 120.4 °N), Lianyungang (34.8 °N, 119.4 °E), Dachen (28.5 °N, 121.9 °E), Xiamen (24.5 °N, 118.1 °E), Dongshan (23.8 °N, 117.5 °E) and Zhelang (22.7 °N, 115.6 °E)) are public and can be obtained and downloaded freely. But, an application is needed if we want the SST data from the rest

stations. Following your suggestion, we add the "Data availability" in the revised paper as follows:

*Data availability. All the four gridded SST analyses used in this study are publicly available and can be downloaded freely from the websites shown in Table 1. The observational in situ SST data from the coastal stations and the coordinates of coastal stations can be obtained from the National Marine Science Data Center, National Science & Technology Resource Sharing Service Platform of China ([http://mds.nmdis.org.cn](http://mds.nmdis.org.cn)). However, the observational in situ SST data from only 9 coastal stations are publicly available. SST data from the rest stations can be obtained after an application to the website.*

**Reply to Referee #1    Report #2**

(1) L 158-159: "directly are not compares pairwise" -> "are not compared directly and pairwise"

**Reply**: We correct "directly are not compares pairwise" to "are not compared directly and pairwise" in the revised paper.

(2) L 217: "ocean front" -> "ocean fronts"

**Reply**: We correct "ocean front" to "ocean fronts" in the revised paper.

(3) L 248: "0.2K" -> The unit of the time component of the EOF is not K, but it is unless since it has been normalised, correct?

**Reply**: We remove "K" in the revised paper.

(4) L 353: "connect with" -> "connected by"

**Reply**: We correct "connect with" to "connected by" in the revised paper.

(5) L 365: "catch" -> "catches"

**Reply**: We correct "connect with" to "connected by" in the revised paper.

(6) L 378: "rleatd" -> "related"

**Reply**: Thanks for pointing this out. We correct "connect with" to "connected by" in the revised paper.

(7) L 381: "the local oceanic effects" -> "local oceanic effects"

**Reply**: We correct "the local oceanic effects" to "the local oceanic effects" in the revised paper.

We made our best to improve the language in the revised manuscript. The changes made will not influence the content and framework of the paper. Here we have not listed the rest language changes but all of them have been marked in yellow color in the revised manuscript. We would like to express our great appreciation to you and reviewers for comments on our paper.

[revised manuscript text omitted]

---

## Author Response (AR3)

**Author's response**

Dear Prof. Belkin:

Thank you very much for the comment on our manuscript. For the revision, we fully considered it and give our response as follows.

Major comments:

1. The authors must address the huge spatial data gap between Stations 10 and 11. A paper about coastal SST with a 6-degree latitudinal gap cannot be accepted. The authors should obtain coastal SST data between 29 N (Station 11) and 35 N (Station 10) and update their analysis.

Reply:

*Currently, more than 100 hydrological stations are operating and monitoring near-shore hydrological conditions. Among these stations, only 26 stations have routinely and continuously recorded since 1960, with a percentage of missing data lower than 4%, Also, these stations have undergone only a few (five and less) and documented relocations. Considering the availability and reliability of data for testing the quality of reanalysis datasets, we chose the 26 stations in our study.*

*However, due to the fact that this area between 29 N (Station 11) and 35 N (Station 10) is a vast muddy coast which is not suitable for hydrological stations, there are only 10 stations (shown in Figure 1). And half of them were built up after 2000s, for example Sheshan, Xiaojishan,Daishan, etc, spanning less than 20 years (Table 1). That is why no station has been chosen between 29 N (Station 11) and 35 N (Station 10).*

*Once again, thanks for your comment which would probably enhance the veracity and credibility of our work. Sorry, but there is not sufficiently data between 29 N and 35 N can support our work.*

*Best regards.*

*Yours sincerely,*

*Yan LI*

[Figure]

Figure 1. Ten hydrological stations between 29 ℕ (Station 11) and 35 ℕ (Station 10)

Table 1 Stations between 29 ℕ (Station 11) and 35 ℕ (Station 10)

[revised manuscript text omitted]